# ENERVERSE: Envisioning Embodied Future Space for Robotics Manipulation

**Siyuan Huang[1,3], Liliang Chen[2] [†], Pengfei Zhou[2], Shengcong Chen[2], Yue Liao[5],**
**Zhengkai Jiang[3], Yue Hu[2], Peng Gao[3], Hongsheng Li[4], Maoqing Yao[2] [‡], Guanghui Ren[2] [‡]**
[1]SJTU [2]AgiBot [3]Shanghai AI Lab [4]CUHK MMLab [5]LV-NUS Lab
**Project Page:** `https://sites.google.com/view/enerverse`
*Email: yaomaoqing@agibot.com, renguanghui@agibot.com* [*]

## Abstract

We introduce ENERVERSE, a generative robotics foundation model that constructs and interprets embodied spaces. ENERVERSE employs a chunk-wise autoregressive video diffusion framework to predict future embodied spaces from instructions, enhanced by a sparse context memory for long-term reasoning. To model the 3D robotics world, we adopt a multi-view video representation, providing rich perspectives to address challenges like motion ambiguity and 3D grounding. Additionally, ENERVERSE-D, a data engine pipeline combining generative modeling with 4D Gaussian Splatting, forms a self-reinforcing data loop to reduce the sim-to-real gap. Leveraging these innovations, ENERVERSE translates 4D world representations into physical actions via a policy head (ENERVERSE-A), achieving state-of-the-art performance in both simulation and real-world tasks. For efficiency, ENERVERSE-A reuses features from the first denoising step and predicts action chunks, achieving about 280 ms per 8-step action chunk on a single RTX 4090. Further video demos, dataset samples could be found in our project page.

## 1 Introduction

Creative AI in vision has achieved significant progress, especially in video generation, where models produce high-quality videos from human instructions [23, 59]. This success highlights the model's spatiotemporal imagination, enabling accurate forecasting of future frames. Similarly, robotic manipulation, a fundamental task in embodied AI, needs accurate predictions of future actions based on language instructions to interact with the physical world. Based on this sharing principle of future space prediction, one natural strategy is to align robotics action prediction with a video generation task to leverage video generation models' imagination capabilities for policy planning. Motivated by this, recent studies [49, 38, 6, 15] have conducted preliminary explorations by fine-tuning general video generation models on robotic manipulation videos to align feature representations with the robotics domain, and predict physical actions. However, such methods [38] often simply adapt general-purpose video generation models to embodied tasks, neglecting the substantial gap between their representation space and the three-dimensional, temporally interconnected robotics environment, thereby hindering accurate action policy prediction. We do not claim a direct monotonic link between pixel-level video quality and control success. Rather, we align the latent space to encode 3D, action-conditioned dynamics so that actions can reliably follow generated trajectories.

To bridge the gap, we propose ENERVERSE, a generative robotics foundation model designed to construct and interpret the robotics 4D (3D with time) world. In ENERVERSE, we employ an autoregressive video diffusion framework that iteratively predicts the embodied future space based on a given instruction. Within this generative paradigm, we define a minimal unit of the future space

---

[*][†] indicates project leader. ‡ indicates corresponding author.

39th Conference on Neural Information Processing Systems (NeurIPS 2025).

as a 'chunk', and the model repeatedly predicts the next chunk to incrementally expand the space. Additionally, to prevent model collapse and enhance the action planning capabilities, we design a sparse context memory mechanism during training. Instead of relying on consecutive memory, this mechanism preserves essential prior content throughout the generation process in a non-redundant manner, theoretically allowing infinite-length sequence generation. While this design achieves stable 2D embodied video generation, it remains insufficient for 3D understanding.

A straightforward solution is directly extending 2D video generation into multi-view video generation and mounting multiple cameras to provide more 3D cues [12]. However, adding cameras increases hardware costs, I/O bandwidth requirements, and system complexity. To circumvent these challenges, we argue that a strong 3D generative prior learned during pre-training can effectively enhance the single-camera setup. During inference, observations from the single camera can be wrapped and rendered into multiple views. To establish this generative prior, we pre-train ENER-VERSE on multi-view video generation and introduce a Multi-View Diffusion Generator Block. This block utilizes ray direction maps to encode camera information and employs temporal attention to seamlessly fuse multi-camera data. Empirically, pre-training with multi-view consistency supplies a 3D prior that benefits even single-camera deployments via rendered auxiliary views.

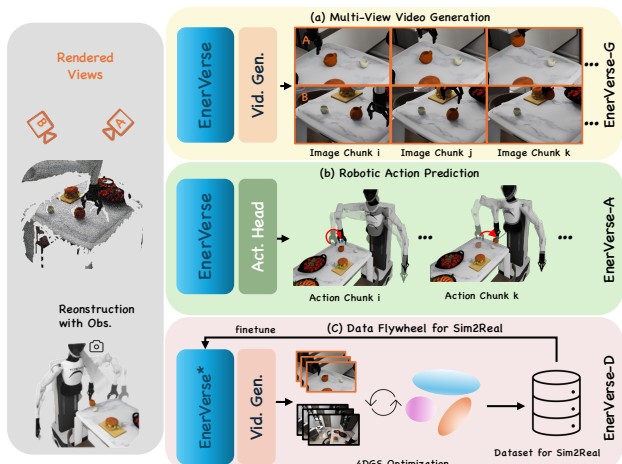

Figure 1: An overview of ENERVERSE. With camera observations, we first obtain a 3D reconstruction via depth warping, then multi rendered images. ENERVERSE (a) connects to a video generator head (ENERVERSE-G) to produce multi-view videos, (b) attaches to a robotic action policy head (ENERVERSE-A) for action prediction, and (c) integrates with 4DGS to form a data flywheel (ENERVERSE-D) for Sim2Real.

Although the pre-training stage requires substantial multi-camera data, acquiring precisely calibrated multi-camera observations with corresponding robotic actions is costly and labor-intensive. While simulators can generate abundant synthetic data, the Sim2Real gap remains a significant challenge. To address this, we propose ENERVERSE-D, a data engine combining a generative model with 4D Gaussian Splatting (4DGS). By leveraging the adaptability of the generative model and the spatial constraints of 4DGS, ENERVERSE-D establishes a data flywheel that narrows the Sim2Real gap.

Building on these designs, ENERVERSE effectively models and interprets the robotic environment in both 3D space and temporal dimensions. With this generative prior, we can directly translate the 4D world (3D spatial with temporal information) representation into physical actions via a policy head, as shown in Fig. 1, allowing the robot to execute task instructions in real-world scenarios. As a result, ENERVERSE-A attains state-of-the-art performance in both simulation and real-world deployments.

The contributions of this work are as follows: (1) A chunk-wise autoregressive diffusion architecture with sparse contextual memory capable of long-term grounding. (2) A multi-view diffusion generator that endows the model with a 3-D spatial prior beneficial under single-camera deployment. (3) A 4DGS-based data flywheel that supplies geometry-consistent multiview training data for robotics.

# 2  ENERVERSE

ENERVERSE comprises several designs, including a chunk-wise autoregressive generation framework and the sparse memory design for embodied future space generation. We additionally integrate a 4DGS to construct a data flywheel, referred to as ENERVERSE-D, and a policy head to generate physical actions, referred as ENERVERSE-A.

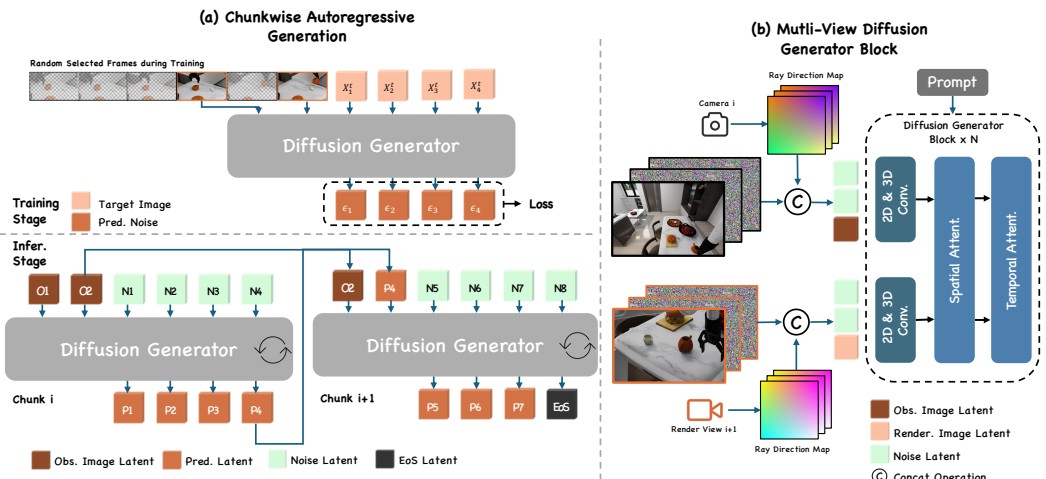

Figure 2: An overview of our chunk-wise autoregressive generation approach and multi-view diffusion generator block. (a) During training, random clean frames from consecutive sequences are combined with noisy frames to predict denoised latents. In inference, newly generated denoised frames become the next clean frames for subsequent steps, iterating until the EoS frame is detected. Only a single view of the autoregressive process is shown for clarity. (b) In the multi-view diffusion generator block, observational frames from Camera $i$ or Rendered View $i + 1$ are encoded with a VAE. Ray direction maps are concatenated with video latents, followed by conv layers and attention mechanisms.

## 2.1 Next Chunk Diffusion

**Chunk-wise Autoregressive Generation.** As shown in Fig. 2, the observed latent sequence is represented as $o_t^{1:K} = [o_t^1, \ldots, o_t^K] \in \mathbb{R}^{K \times H \times W \times C}$, encoded by a pre-trained Variational Autoencoder (VAE). Here, $K$ denotes the number of observed frames, $H \times W$ represents the spatial resolution, $C$ is the number of channels, and $t$ is the denoising step. Similarly, the latent representation of the rendered image is given by $r_t^{1:J} \in \mathbb{R}^{J \times H \times W \times C}$. For simplicity, we treat $r$ as a special case of $o$. The predicted latent sequence is denoted as $z_t^{1:M} = [z_t^1, \ldots, z_t^M] \in \mathbb{R}^{M \times H \times W \times C}$. The goal is to develop a video diffusion model that generates these predicted latents conditioned on $o_0^{1:K}$ and a textual prompt $c$, following the conditional probability: $p_\theta(z_t^{1:M} \mid c, o_t^{1:K})$. Here, $\theta$ represents the parameters of the denoising network, which is defined as $\epsilon_\theta(z_t^{1:M}, c, o_t^{1:K}, t)$. $c$ is encoded by a frozen T5 encoder and then projected with an MLP. The network is trained to predict the ground truth noise $\epsilon$ from the noisy frame targets by optimizing the loss function:

$$\min_\theta \mathbb{E}_{t, \mathbf{z} \sim z_{\text{data}}, \epsilon \sim \mathcal{N}(\mathbf{0}, \mathbf{I})} \left\| \epsilon - \epsilon_\theta \left( z_t^{1:M}, o_t^{1:K}, t \right) \right\|_2^2,$$

where $\epsilon$ is the sampled ground truth noise, and $\theta$ denotes the learnable parameters. We follow [41] to implement the v-prediction. Instead of predicting the noise $\epsilon_t$, the model predicts $v_t$, defined as: $v_t = \alpha_t \epsilon_t - \sigma_t x_0$. Here, $\alpha_t = \sqrt{\bar{\alpha}_t}$ (signal scale) and $\sigma_t = \sqrt{1 - \alpha_t^2}$ (noise scale), consistent with the forward process equation $x_t = \alpha_t x_0 + \sigma_t \epsilon_t$. After training, the denoised data $\mathbf{z}_0$ can be derived from random noise $\mathbf{z}_T$ through iterative denoising.

During inference, the diffusion generator takes both clean and noisy frames as input to produce $M$ denoised frames. The newly generated frames serve as clean inputs for subsequent iterations, and this process repeats until detecting a predefined End-of-Sequence (EOS) frame. As the diffusion generation operates on latent frames, the L1 distance of each frame to the EOS is computed. If this distance falls below a predefined threshold, generation is terminated. In practice, this threshold-based EOS detection is highly effective.

**Sparse Memory Mechanism**. Instead of the conventional approach of using consecutive frames as the clean frame context for chunk prediction during training, we propose using sparsely sampled frames as the clean frame context. This approach leverages the redundancy often present in video data, allowing approximately 80% of frames to be discarded without compromising training effectiveness. Additionally, the high frame-dropping ratio enhances the model's robustness, particularly in handling out-of-distribution (OOD) scenarios such as covariant shift problems commonly encountered in the

robot learning domain. From a representation learning perspective, this randomized sampling strategy promotes a deeper understanding of chunk prediction, potentially outperforming methods that rely on continuous frames.

During inference, clean frames are derived from observed or rendered frames and denoised using a sliding window approach. This technique ensures a smooth transition between observed and generated frames while improving efficiency and reducing GPU memory consumption.

## 2.2 4D Embodied Space Generation

Single-view video generation struggles to recover accurate 3D structure and resolve occlusions, which is critical for embodied manipulation. We therefore extend the next-chunk diffusion model in Sec. 2.1 into a multi-view video generator that is conditioned on camera geometry and learns cross-view consistency end-to-end. Given a task prompt $c$, $m$-view observations $O_{1:K} \in \mathbb{R}^{K \times m \times H \times W \times 3}$ with per-view camera intrinsics and extrinsics, we encode each frame into VAE latents and directly predict future multi-view latents $\mathbf{z}_{1:M} \in \mathbb{R}^{M \times m \times H \times W \times C}$, as illustrated in Fig. 2. To make the representation view-aware, we compute a per-view ray-direction map that encodes the camera intrinsics and extrinsics and concatenate it channel-wise with the image latents before the diffusion backbone, following ray-based conditioning [7, 40]. Cross-view geometric coherence is modeled using attention along the view dimension: we reshape features to attend across views at corresponding spatial locations and apply spatial attention that preserves pixel-to-ray alignment. Temporal attention is applied along the time dimension to capture scene dynamics. During training, simulator data provide ground-truth camera parameters; for real-world videos we use estimated extrinsics relative to the robot base. At inference, we use the available extrinsics; when depth is available, we optionally depth-warp observed frames to synthesize auxiliary rendered frames that better match the multi-view training conditions.

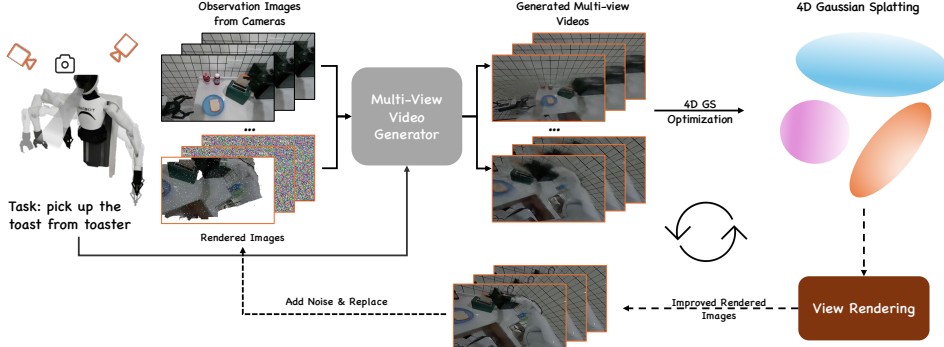

Figure 3: The pipeline for ENERVERSE as a data engine. Observation images from multiple cameras and rendered images are processed by the multi-view video generator to produce denoised videos. These videos, along with their camera poses, are used in 4DGS for 4D scene reconstruction. The reconstructed 3D content is rendered to generate high-precision images. These high-quality rendered images are iteratively refined and fed back into the pipeline.

**Real-World Data Flywheels.** To reduce reliance on costly fully calibrated multi-camera capture and to narrow sim-to-real gaps, we introduce an offline data flywheel, e.g., ENERVERSE-D, that leverages sparse real observations to bootstrap geometry-consistent multi-view videos and progressively reduce the gap, as shown in Fig. 3. After pretraining the 4D base model (EnerVerse-G) as above, we fine-tune it to accept *sparse* multi-view inputs in the 4D latent space $\mathbb{R}^{C \times V \times T \times H \times W}$. Specifically, among $m$ views, at least $n \ll m$ robot-mounted cameras provide complete observation sequences; for these observed views we skip noise injection and use their clean latents as conditioning, while we apply the standard noisy-to-denoised diffusion to unobserved target views. Given sparse observations (e.g., one full video), their camera parameters, and a task prompt, the model produces denoised predictions for the missing views. We then reconstruct a 4D scene using 4D Gaussian Splatting (4DGS) from the union of observed and generated multi-view videos and their poses. The 4DGS representation is rendered to all target views to obtain higher-fidelity, geometry-consistent frames. These renders can be re-noised and fed back through the multi-view generator, followed by another 4DGS optimization step, yielding an iterative loop that progressively reduces noise, improves reconstruction accuracy,

and tightens cross-view consistency. As the loop accumulates real multi-view episodes, we further fine-tune the multi-view generator on the collected data.

## 2.3 From 4D Embodied Space to Physical Action

We further integrate a policy head into the diffusion generator, enabling the generation of actions after the extensive pretraining of future space generation. The policy operates on a compact visual latent $E$ extracted from the middle block of the UNet backbone at the first denoising step (the noisiest step), which reduces computation while retaining rich task-relevant cues. Visual inputs may be captured RGB frames or rendered views, as shown in Fig 4; both are encoded by the shared video backbone. The action head $h_\theta$ is a stack of DiT blocks followed by a linear projection to the action space. We predict action chunks [58] for temporal consistency. Let $a_{t:t+\tau-1} \in \mathbb{R}^{\tau \times d}$ denote a chunk of actions with $d = 7$ (delta position, rotation, and gripper openness). The denoising model $f_\theta$ estimates clean actions from noisy inputs using DDPM-style training:

$$a_{t:t+\tau-1}^0 \leftarrow f_\theta(c, o_t, a_{t:t+\tau-1}^k, k) \ = \ h_\theta(E, a_{t:t+\tau-1}^k, k),$$

where $E$ is the cached visual latent from ENERVERSE-G and $k \in \{1, \ldots, K\}$ is the diffusion step. The training objective minimizes denoising MSE.

At inference, we compute $E$ once by passing $(c, o_t)$ through the video diffusion backbone and cache it across the action denoising steps. The action head then iteratively denoises from $a^K$ to $a^0$ for the current chunk. We use per-view, per-frame decoding for visuals, but policy conditioning is view-aggregated via $E$ (mean over spatial dimensions within the UNet middle block). We provide more details in the Appendix.

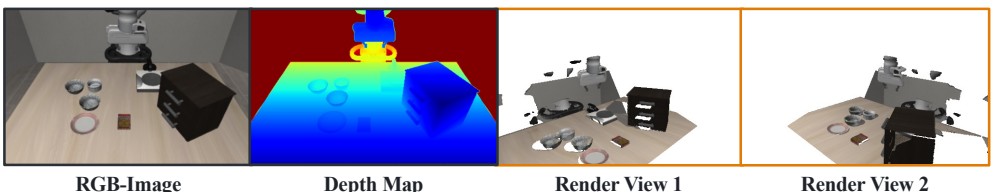

| RGB-Image | Depth Map | Render View 1 | Render View 2 |

Figure 4: Render View 1 and Render View 2 are generated by rendering from a point cloud reconstructed from RGB-Image 1 using depth wrapping. The render views correspond to camera views obtained by rotating the RGB camera view around the Z-axis by $\pm 30°$.

## 3 Experiments

To demonstrate the effectiveness of proposed method, we evaluate ENERVERSE in two different domains, e.g. video generation quality and robotic policy performance.

### 3.1 Experiment Settings

**Training Data:** We selected several public datasets characterized by well-defined task logic, including RT-1 [4], Taco-Play [39], ManiSkill [14], BridgeV1 [46], LanguageTable [27], and RoboTurk [28] for pretraining. Furthermore, we constructed a dataset containing multi-view video ground truths using the Isaac Sim simulator [31]. The detailed dataset statistics could be found in Appendix. During pretraining , only video frames were utilized for video generation training. For the policy planning task, fine-tuning with a limited quantity of demonstration data from specific scenarios proved sufficient to attain state-of-the-art performance. To mitigate domain gaps encountered when training with heterogeneous data, we employed domain embeddings inspired by [47]. Specifically, distinct domain embeddings were allocated to each sub-dataset. In subsequent space generation and policy planning, these embeddings were integrated with the diffusion timestep embeddings prior to input into the diffusion model. This methodology effectively alleviated conflicts arising from discrepancies in entities, task types, and visual styles.

**Model Details.** Our model is conducted based on UNet-based Video Diffusion Models (VDM) [53], and can be easily adapted to DiT [32] architectures. And the image decoding occurs per-view and per-frame. In our experiments on generating embodied future spaces, we identified that chunk size significantly influences model performance. Comparative analyses utilizing chunk sizes of 1, 4, 8, and 16 revealed that the model exhibited optimal robustness when employing a chunk size of 8.

Following the methodology outlined in [5], we introduced corruptive noise to the frames within the memory context. To alleviate degradation in autoregressive generation, the intensity of this noise was modulated in a cosine-related manner relative to the distance from the current moment. After pretraining the multiview video generation models, we performed a generation learning with the action videos to achieve both visual and spatial adaptation. Subsequently, we fine-tuned the action policy head using action trajectories. Following this, we fine-tune the action policy head using the action trajectories. The action head adopts the Diffusion Policy (DP) architecture [10], with a total of 190M parameters. For the DP head's condition, we utilized features extracted from the middle block of the UNet during the first denoising step and calculated the mean across the spatial dimensions. This resulted in a final shape of $T \times C$, where $T$ is the video length and $C$ is the channel number. Rendered images have a resolution of $512 \times 320$, and the action head predicts the delta pose. For ENERVERSE-D, we integrate 4D Gaussian Splatting using the official implementation [50], with depth-based initialization when available and a deformation depth of 1.

## 3.2 Comparison Results

**Embodied Future Space Generation.** Following AVID [38], we assess video generation quality utilizing the RT-1 [4] dataset. To create a comparable baseline, we fine-tune DynamicCrafter on the RT-1 dataset and run inference iteratively with FreeNoise [34] to enable long video generation(DC-FN). For evaluation, we generate 200 synthetic videos with varied lengths by conditioning the models on the initial frame and task instructions, subsequently comparing the generated videos against the ground truth using standard metrics such as PSNR and FVD. However, while these metrics primarily evaluate visual quality, embodied tasks necessitate additional considerations, including semantic alignment with instructions, workspace consistency across frames, and motion continuity. To address these higher-order aspects, we execute a user study involving robotics experts, assessing the generated videos based on semantic accuracy, frame consistency, and motion continuity.

| Method | Atomic Task | | | | | | Long Task |
| --- | --- | --- | --- | --- | --- | --- | --- |
| | PSNR↑ | FVD↓ | Quality↑ | Seman.↑ | Consist.↑ | Continuity↑ | Ability |
| DC-FN | 25.42 | 445.94 | 54 | 97 | **92** | 80 | × |
| ENERVERSE | **26.1** | **404.65** | **59** | 97 | 89 | **90** | ✓ |

Table 1: Performance comparison between DynamiCrafter (FN) and our proposed approach across Atomic Task metrics (Quantitative Comparison and User Study) and Long Task ability. The proposed method outperforms DynamiCrafter (FN) in most metrics, demonstrating its effectiveness in video generation and task performance.

Tab. 1 illustrates that our method substantially outperforms DynamicCrafter (FN) in both quantitative and qualitative evaluations. In terms of quantitative metrics, our approach achieves a higher PSNR and a lower FVD. These findings indicate that our method produces videos of superior visual quality and enhanced temporal dynamics. In the user study, our method secures a higher quality score and exceeds DynamicCrafter in motion continuity, which is essential for robotic manipulation tasks. Although both methods attain equivalent semantic accuracy, this suggests that our approach effectively preserves instruction alignment while delivering superior overall performance. Moreover, our method uniquely accommodates long tasks, as evidenced by its successful execution of long-range manipulation scenarios, whereas DynamicCrafter falters in this domain. We also provide a qualitative comparison in Fig. 5.

**Multi-View Generation Consistency**. In this section, we qualitatively demonstrate the capability of ENERVERSE to generate multi-view videos of the same scene while ensuring consistency across views. Furthermore, each view attains high-quality image generation, thereby highlighting the robustness of our approach. As shown in Fig. 6, ENERVERSE could generate high-quality multi-view videos in both simulator and real-world settings.

**Robotic Policy Evaluation on LIBERO** Following the evaluation protocol in OpenVLA [22], we evaluate robotic policies using the LIBERO [26] benchmark, which consists of four distinct task suites: LIBERO-Spatial, LIBERO-Object, LIBERO-Goal, and LIBERO-Long. Each suite contains 10 tasks, each with 50 human demonstrations. For each task suite, a separate policy model is fine-tuned. We compare our method against five baselines: *Diffusion Policy* [10], a direct action learning policy trained from scratch; *Octo* [44], a transformer-based policy model fine-tuned on the target dataset;

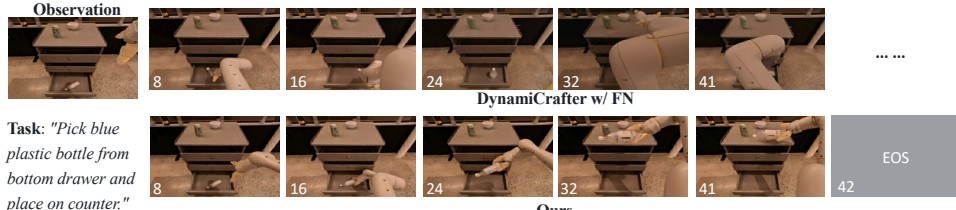

Figure 5: Qualitative comparison for single view video generation between ENERVERSE and DynamiCrafter(FN) on RT-1 dataset. Since ENERVERSE predict EOS frame at 42th frame for this task, we visualize up-to 42th frame sampled from both generated sequence. The sequences generated by DynamiCrafter(FN) did not maintain the logic and produce many hallucinations as the sequence grew. In contrast, the sequence generated by ENERVERSE was logically coherent, continuously and completely generating the future space of the entire task, and accurately predicting the EOS frame.

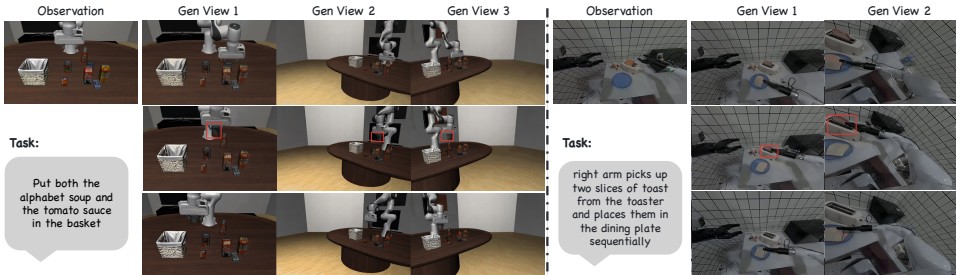

Figure 6: Qualitative results for multi view generation on LIBERO (left) and real-world manipulation data (right). The first generated view position is overlapped with a static mounted RGB camera and others are manually set. The consistency of objects across views is highlighted by a red rectangle.

*OpenVLA*, a 7B vision-language-action (VLA) model fine-tuned on the target dataset; *MDT* [37],a diffusion transformer-based policy with an auxiliary MAE loss; *MAIL* [19], a policy model with Mamba [13] in an encoder-decoder structure. Besides, we provide the results of MAIL with two S-RGB input with their official implementation. For evaluation, all models are tested across tasks using 50 rollouts per task, with results averaged over three random seeds. Experiments with ENERVERSE-A are conducted under three setups: a single RGB image, and when RGB-D is available, 1 RGB image with 1 rendered view, and 1 RGB image with 2 rendered views, as shown in Fig. 4. The abbreviations denote different input modalities: S-RGB for Static RGB, G-RGB for Gripper RGB, S-RGBD for Static RGB-D, G-RGBD for Gripper RGB-D, P for proprioceptive arm position.

| Model | Visual Input | Spatial | Object | Goal | Long | Avg. |
|---|---|---|---|---|---|---|
| **Diffusion Policy** | S-RGB | 78.3 | 92.5 | 68.3 | 50.5 | 72.4 |
| **Octo** | S-RGB | 78.9 | 85.7 | **84.6** | 51.1 | 75.1 |
| **OpenVLA** | S-RGB | 84.7 | 88.4 | 79.2 | 53.7 | 76.5 |
| **MDT** | S-RGB,G-RGB | 78.5 | 87.5 | 73.5 | 64.8 | 76.1 |
| **MAIL** | S-RGB,G-RGB | 74.3 | 90.1 | 81.8 | 78.6 | 81.2 |
| **MAIL** | S-RGB,S-RGB | 76.0 | 90.0 | 82.0 | 78.0 | 81.5 |
| **ENERVERSE** | S-RGB | **92.1** | 93.2 | 78.1 | **73.0** | **84.1** |
| **ENERVERSE** | S-RGBD → RGB with 1 Render | **93** | 95.0 | 81.0 | 73.0 | 85.5 |
| **ENERVERSE** | S-RGBD → RGB with 2 Render | **91.2** | **97.7** | **85.0** | **80.0** | **88.5** |

Table 2: Evaluation results on the LIBERO benchmark across four task suites.

As shown in Tab. 2, ENERVERSE achieves state-of-the-art performance across the LIBERO benchmark, significantly surpassing all baselines. With single S-RGB input, it achieves an average score of 84.1, outperforming strong baselines.

**Robotic Policy Evaluation on CALVIN** CALVIN [29] is an open-source simulated benchmark designed for learning long-horizon tasks. It consists of four distinct scenes (A, B, C, and D) and introduces the ABC→D evaluation protocol, where models are trained on environments A, B, and C and evaluated on environment D. The primary evaluation metric is the success sequence length, which measures the ability to complete five consecutive subtasks within a sequence. Notably, CALVIN's training data is trajectory-based, whereas inference requires performing sequential tasks without explicit task transition signals. This discrepancy between training and inference introduces challenges, as our model relies on memory. Nevertheless, we strictly adhere to the evaluation protocol and do not reset the memory when a new task begins, making policy inference more demanding. This limitation does not affect other models, as they do not utilize memory. Despite these challenges and limited input signals, our model achieves competitive performance, as shown in Table 3.

| Method | Input | 1 | 2 | 3 | 4 | 5 | Avg. Len. |
|---|---|---|---|---|---|---|---|
| **RoboFlamingo** [25] | S-RGB, G-RGB | 82.4 | 61.9 | 46.6 | 33.1 | 23.5 | 2.47 |
| **GR-1** [51] | S-RGB, G-RGB, P | 85.4 | 71.2 | 59.6 | 49.7 | 40.1 | 3.06 |
| **3D Diffuser** [20] | S-RGBD, G-RGBD, P | 92.2 | 78.7 | 63.9 | 51.2 | 41.2 | 3.27 |
| **SUSIE** [2] | S-RGB | 87 | 69.0 | 49.0 | 38.0 | 26.0 | 2.69 |
| **ENERVERSE** | S-RGB | 90.8 | 73.0 | 57.3 | 43.7 | 35.6 | 3.00 |

Table 3: The comparisons with state-of-the-art approaches on Calvin (ABC → D).

### 3.3 Further Studies

In this section, we explore several key design choices for ENERVERSE. First, we examine the significance of the proposed sparse memory mechanism, which plays a critical role in both policy learning and video generation. Second, we discuss the training strategy utilized in ENERVERSE. Third, we analyze the alignment between the predicted action spaces and visual spaces through attention map analysis. Finally, we introduce the real-world experiment setup.

**Effectiveness of Sparse Memory Mechanism**. We evaluate the effectiveness of our sparse memory mechanism in both policy learning and video generation. The evaluation is conducted on the LIBERO-Long task suite, as this suite involves significantly longer task execution steps, requiring the policy to exhibit strong long-range memory and task reasoning capabilities. The evaluation is performed with a single visual input. As shown in Tab. 4, the absence of the *sparse memory* results in significant performance degradation, with the policy achieving only 30.8 compared to 73 when the sparse memory mechanism is applied. Similarly, Fig. 7 demonstrates that when the video generator operates without sparse memory, the model experiences unexpected collapse and fails to recover in out-of-distribution (OOD) scenarios. In contrast, the sparse memory mechanism ensures robust performance while also saving computational resources.

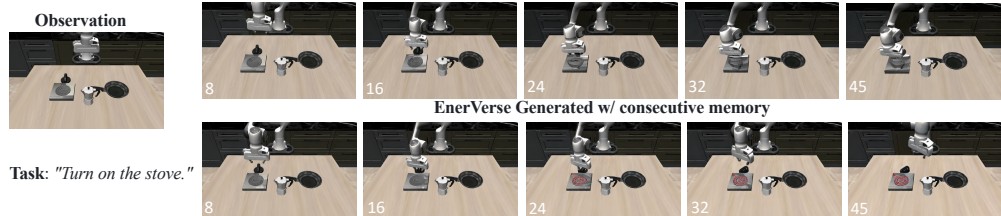

Figure 7: Ablation results for context memory mechanism in video generation. Providing history information to the generation model with consecutive context (first line) often leads to unexpected model collapse while the model with sparse memory (second line) shows robust performance and save mush computing resources.

**Training Strategy Analysis**. To analyze the impact of different training strategies on robotic policy learning, we trained four robotic policies on the LIBERO-Spatial task suite using the following approaches: (1) training the entire ENERVERSE from scratch using only policy loss optimization; (2) training the entire ENERVERSE as in (1) but initialized with pretrained weights from a general video generator, e.g. DynamiCrafter(DC) [53], which is trained with the general natural videos; (3) co-training ENERVERSE by optimizing both the robotic policy action loss and the video generation

| Setup | w/o Sparse Memory | w Sparse Memory |
|---|---|---|
| LIBERO-Long-SV | 30.8 | 73 |

Table 4: Sparse Memory analysis on LIBERO-Long.

loss simultaneously; and (4) the default two-stage training strategy, where the video generator is pretrained first, followed by fine-tuning ENERVERSE using only robotic policy loss optimization.

| Strategy | All-Scratch | With DC Pretrain. | One-Stage Co-Train | Two-Stage Finetune |
|---|---|---|---|---|
| **LIBERO-Spatial** | Failed | 79 | 86.3 | **92.1** |

Table 5: Performance comparison of different training strategies on the LIBERO-Spatial task suite.

As shown in Tab. 5, training ENERVERSE from scratch without loading pretrained weights failed to converge, underscoring the importance of robust initialization. Another possible reason for this failure could be the relatively limited training data compared to the number of network parameters. Initializing with pretrained weights improved performance (79), while jointly optimizing the policy loss and video generation loss in a one-stage co-training setup further increased performance to 86.3. This demonstrates that the video generation task enhances policy learning. Our default Two-Stage Fine-tuning strategy, which involves pretraining the video generator followed by fine-tuning ENERVERSE with policy loss optimization, achieved the best performance.

**Effectiveness of Additional Rendered Views**. With the expressive pretrained generative space prior, our method using a single S-RGB camera already achieves SOTA performance. When a single S-RGBD camera is available, we can incorporate additional rendered views, with the original RGB image as input to the model. These additional views not only bring the input setting closer to the training configuration but also enable the policy model to better leverage the pretrained generative space prior. Notably, the configuration `RGB with 1 Render` outperforms MAIL's `2 S-RGB` setup, both in terms of overall performance and gains compared to a single RGB input, demonstrating that the performance improvement is not solely due to the additional visual inputs. Incorporating `RGB with 2 Render` yields even greater gains by mitigating occlusions, as illustrated in Fig. 4.

**3D Video vs. 4D Space for Robotics**. We provide a direct comparison between attaching a diffusion policy head to a base single-view video generator (DynamiCrafter) and our 4D variant (EnerVerse-A). As shown in Table 6, the base video model underperforms substantially (79.0) relative to ENERVERSE-A (92.1), highlighting the benefits of 4D extensions. We hypothesize that the cross-view consistency learned during multi-view pretraining provides stronger geometric priors, which help the model better understand spatial relationships and occlusions. Further, even when tested with single RGB-D inputs, ENERVERSE benefits from additional rendered views at inference. These findings underscore the ability to incorporate additional rendered views at test time further enhances performance, showcasing the practicality and effectiveness of our approach.

| Model | Multi-view Pre-Train | Input at Test | SR |
|---|---|---|---|
| DynamiCrafter + DP | No | S-RGB | 79.0 |
| EnerVerse-A | Yes | S-RGB | 92.1 |
| EnerVerse-A | Yes | S-RGB with 1 Render | 93.0 |

Table 6: Direct comparison between a base single-view video generator with a policy head and our approach on LIBERO-Spatial. Multi-view pretraining substantially improves SR even with single-view inputs.

**Real-World Experiments.** To evaluate the manipulation capabilities of ENERVERSE-A, we conducted real-world experiments using commercial robotics on the tasks of `Block Placing`, `Plastic Objects Sorting`, and `Fruit Sorting`. For further details, please refer to Appendices A.

**More Discussions and Experiments.** We provide additional discussions and experiments in the appendices: pretrained model performance (Appendices C), alignment between action and visual spaces (Appendices D), robustness against OOD samples (Appendices E), model architecture and computational overhead (Appendices F and G), and visual samples validating the data engine's effectiveness (Appendices I).

# 4 Related Works

**Video Generation Models.** Diffusion-based video generation models have made notable progress, especially in text-to-video (T2V) generation [3, 42]. Early T2V approaches [56, 9, 35, 16] build on text-to-image (T2I) priors by introducing temporal modules trained on video data. Dynamic-Crafter [53] reuses motion priors from T2V diffusion models in an image-to-video (I2V) context. Recent works [23, 59, 1] explores replacing U-Nets with Diffusion Transformer (DiT) [33], and [36] uses the multi-camera poses information to extend the video generation into the 3D world modelling. Other studies [11] incorporate causal mechanisms to generate longer sequences or extend video-generation models into world modeling by forecasting future states [17, 5, 48]. In this paper, we mainly adopt DynamicCrafter as our base I2V framework due to its open-source availability and widespread use. We also ensure compatibility with modern DiT architectures, although that is not our main focus here.

**Video Pretraining for Robotics.** GR-2 [6] presents a generalizable robot manipulation framework that pretrains on large-scale internet videos, then fine-tunes on both video generation and action prediction for robotic trajectories. LAPA [55] uses non-robot action videos for representation learning, mapping discrete latent actions (via VQ-VAE) to robotic manipulation tasks through a VLA model. SEER [45] further explores inverse dynamics pretraining to boost performance. AVID [38] employs DynamicCrafter [53] as its foundation, using an adapter for the robotics domain. VidMan [49], based on OpenSora [59], focuses on environment prediction before action generation but is limited to 2D image space. In contrast, we propose generating long-sequence futures via a novel data generation engine, capturing richer motion information vital for robotics.

**4D Generation.** Recent progress [8] allows reconstruction of dynamic scenes from 2D videos using 3D Gaussian Splatting (GS) [21] and Neural Radiance Fields (NeRF) [30]. Prior approaches approximate the spatio-temporal 4D volume with sets of 4D Gaussians [54], jointly optimizing geometry and motion in canonical space [50]. More recent advancements [24] employ customized sampling for multi-view video diffusion models, particularly for single dynamic objects. DimensionX [43] leverages multiple LoRAs [18] for diverse camera motions, while Cat4D [52] uses a single multi-view diffusion model to generate videos for dynamic 3D reconstruction. By contrast, our method produces videos from a multi-views tailored for robotic manipulation tasks. In our offline data flywheel stage, GS complements video generation models to mitigate the sim-to-real gap, enhancing geometric consistency and reducing hallucinations from the generative models.

# 5 Conclusions

In conclusion, ENERVERSE is a generative robotics foundation model that tackles multi-view video generation and long-range policy execution by modeling embodied future spaces. With sparse contextual memory and chunkwise autoregressive architecture, ENERVERSE enhances spatial reasoning and task adaptability. The ENERVERSE-D pipeline, combining generative modeling with 4DGS, bridges the sim-to-real gap, reducing reliance on real-world data. Integrated with a policy head, ENERVERSE-A achieves state-of-the-art performance in manipulation tasks, both in the simulator environment and the real-world settings.

**Limitations.** Due to the high dynamics and rich object interactions in robotics tasks, video generation models inevitably produce artifacts, as discussed in App. B. However, advancements in the video modeling community are expected to improve this. Additionally, while we provide an initial attention map analysis, further exploration is needed to better understand how video generation guides action policy learning. Finally, the current rendered views are derived from heuristically set camera poses, which may not be optimal. Integrating Next-Best View methods [57] could address this limitation.

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

# A Real-World Robotic Experiments

To evaluate the manipulation capabilities of ENERVERSE-A, we conducted real-world experiments. The robot is instructed to place blocks into designated compartments of a foam worktable, requiring accuracy due to the tight fit and visual similarity between the foam and table, as shown in Figure 8.

Compared with the general "Pick and Place" task, this task has additional challenges:

- The robot must follow natural language instructions, such as "Row One, Column Two," to identify the required compartment.
- The compartments are only slightly larger than the magnet blocks, transforming the pick-and-place task into a highly precise "insertion" operation.
- The magnet blocks are relatively heavy, requiring the robot gripper to grasp near the center of the block to ensure stability during manipulation.

Correspondingly, we define four evaluation metrics:

- **Grasp**: Indicates whether the robotic gripper holds the suitable part of the block and transfers it stably during manipulation. It has binary values: 0 for failure, 1 for success.
- **Place**: Determines whether the robot places the block into a possible compartment. A score of 0 indicates failure, 1 indicates a perfect placement, and 0.5 indicates that the block has some collisions with the foam during manipulation.
- **Instruction Following**: Evaluates whether the robot places the block into the desired compartment as instructed. It has binary values: 0 for failure, 1 for success.

The overall **Success** is calculated as the product of the individual factors. The policy was executed five times for each compartment, and the average scores are presented in Table 7. ENERVERSE-A demonstrates strong performance in most target positions. However, it fails to handle positions $(3, 2)$ and $(3, 3)$. We hypothesize that this limitation arises because these positions are located near the boundary of the robot's action space, making them challenging to reach. We provide the OpenVLA [22] results. Our method outperforms OpenVLA in both grasp and place subtasks, demonstrating superior spatial understanding. The place subtask, in particular, is challenging due to compartments being only slightly larger than the blocks, requiring precise spatial understanding and target localization, which highlights the benefits of our method's 4D space prior. However, OpenVLA shows better instruction-following ability, likely due to its LLM part (our CLIP text encoder). Demonstration videos are provided in the supplementary materials.

In addition to the block placement task, we conducted experiments on sorting transparent plastic objects and fruit sorting. Demonstration videos for these experiments are also included in the supplementary materials.

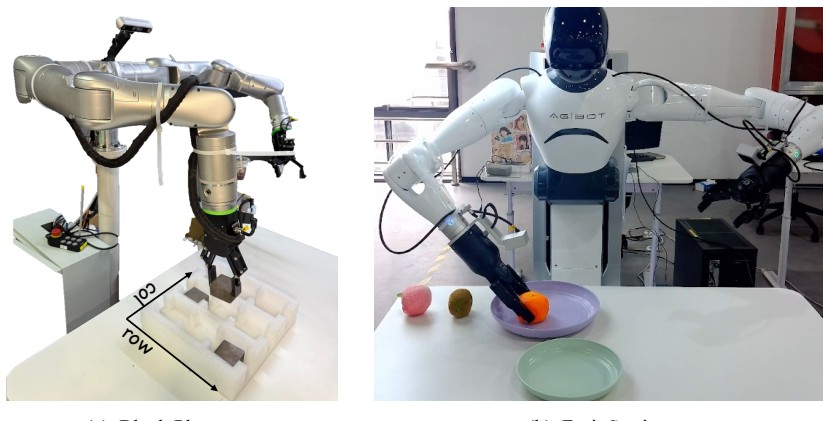

(a) Block Placement      (b) Fruit Sorting

Figure 8: Real-world experimental setup.

| Target Position | Grasp | Place | Ins. Following | Success |
|:---:|:---:|:---:|:---:|:---:|
| (1,1) | 1 | 1 | 1 | 1 |
| (1,2) | 1 | 1 | 1 | 1 |
| (1,3) | 1 | 0.8 | 1 | 0.8 |
| (2,1) | 1 | 0.7 | 1 | 0.7 |
| (2,2) | 1 | 1 | 1 | 1 |
| (2,3) | 1 | 0.8 | 1 | 0.8 |
| (3,1) | 1 | 0.7 | 1 | 0.7 |
| (3,2) | 1 | 1 | 0 | 0 |
| (3,3) | 1 | 1 | 0 | 0 |
| OpenVLA-Avg | 0.89 | 0.61 | 0.96 | 0.61 |
| ENERVERSE-Avg | 1.0 | 0.89 | 0.78 | 0.67 |

Table 7: Performance of the robotic system in placing blocks into designated compartments. The task demands high precision due to the tight fit and visual similarity between the foam and table.

# B   Further Discussions on the Tasks Types and Video Quality in the Real-World Settings.

Integrating physical knowledge, such as kinematics and dynamics, into generative models remains a significant challenge, particularly for complex robotic manipulation tasks [60]. However, we believe that large-scale, high-quality pretraining data can significantly enhance physical modeling capabilities, including tasks involving deformable object manipulation. For instance, we provide video generation results for a cloth-folding task in Fig. 9. Our approach is not limited to simple pick-and-place tasks. It is capable of modeling tasks that require kinematic constraints, such as articulation (e.g., turning a button or opening a drawer). Additional video results are provided in the supplementary materials.

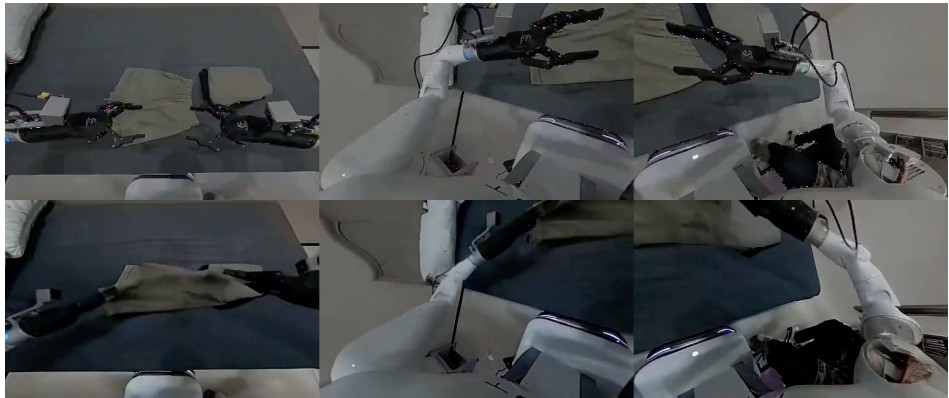

Figure 9: Generation Results on the Cloth Folding Task.

While visual artifacts in the generated videos (e.g., surface penetration or snappy transitions) are present, these imperfections have minimal impact on robotic task execution. In our framework, the generated videos primarily serve as a 4D spatiotemporal prior, which is further refined during fine-tuning. This is supported by our real-world robot experiments, where task performance remains robust despite the presence of these artifacts.

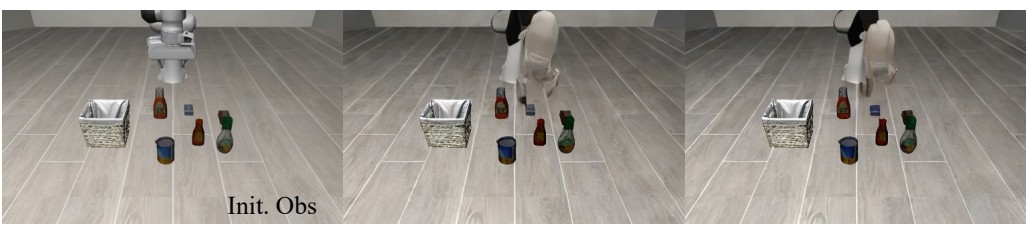

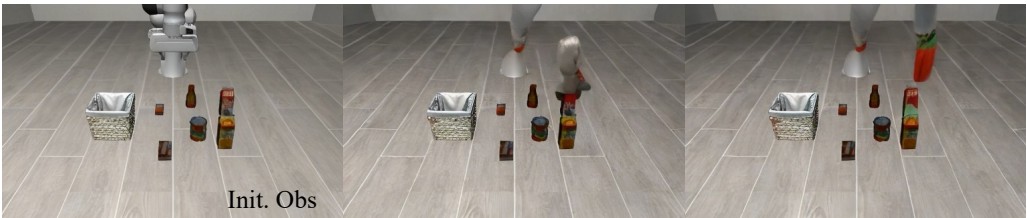

Figure 10: Generation Results with Pre-Trained Model on LIBERO.

## C   Discussion on the Pretrained Model Performance

We present the pretrained model's generation results on the LIBERO-Object split in Fig. 10. The generated videos exhibit significant artifacts, with the scenes collapsing after several frames. We attribute this issue to the domain gap between the LIBERO dataset and the datasets used for pretraining.

Additionally, we directly fine-tuned the pretrained model on LIBERO-Object actions without adapting it to the LIBERO video generation task. As shown in Table 8, this approach results in substantial performance degradation for the final policy model.

| Method | LIBERO-Object |
|---|---|
| ENERVERSE | 93.2 |
| ENERVERSE w/o Gen-Adaption | 85 |

Table 8: Comparison of policy performance w/wo LIBERO video generation task adaption.

## D   Attention Map Analysis

. To further analyze the alignment between the predicted action space and the visual space, including the visual observations cached by our Sparse Memory Mechanism and the generated future space, we visualized the attention maps from the first several layers of the Cross-Attention Block in our policy head.

Fig. 11 illustrates attention maps from different heads and layers, showcasing the model's hierarchical focus and the impact of our proposed embodied future space generation in facilitating robust action prediction. In Fig. 11(a), attention is distributed almost entirely across the future space, reflecting the model's ability to leverage sparse memory conditions and generated predictions from the outset. In contrast, Fig. 11(d) shows the attention sharply focused on the sparse memory space, with minimal reliance on the generated future space, indicating that the model has transitioned to memory-based reasoning. Interestingly, Figures 11(c,e) demonstrate that the model effectively integrates information from both the sparse memory space and the predicted future space. Moreover, these attention maps reveal that earlier decision steps tend to prioritize sparse memory, while later action steps shift focus to the generated future space. These results validate that our generative pretraining effectively enhances the model's ability to integrate temporal information, align predicted actions with future visual contexts, and make robust decisions.

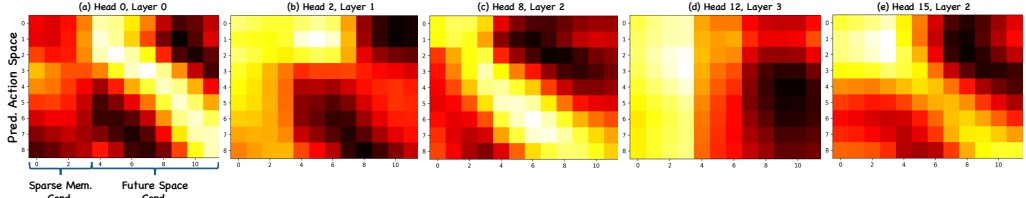

Figure 11: Attention maps from different heads and layers of the model. The y-axis (Query) represents the predicted action space (8 steps), while the x-axis (Key-Value) spans Sparse Memory (first 4 columns) and predicted future space (last 8 columns). Bright yellow indicates high attention, showing how the model focuses on memory (left) and future predictions (right) when generating actions.

## E    Robustness against OOD Samples

To evaluate generalization on out-of-distribution (OOD) samples, we designed three experiments:

- Changing the floor texture for unseen scenes.
- Altering container textures for unseen containers.
- Training on all LIBERO splits ("train-all, test-all") and evaluating on each split simultaneously.

The first two experiments required no retraining and were conducted on the LIBERO-Object split. As shown in Table 9, our model demonstrated strong generalization and robustness. For the "train-all, test-all" experiment, the performance (87.63 Avg) improved compared to single-split training (84.1 Avg). We attribute this improvement to shared textures and spatial layouts across splits, which enable better learning from the larger mixed dataset.

| Method | Seen | Uns. Scene Texture | Delta | Uns. Cont. Texture | Delta |
|---|---|---|---|---|---|
| OpenVLA (S-RGB) | 88.4 | 64.9 | 23.5 | 82 | 6.4 |
| Ours (S-RGB) | 93.2 | 93.1 | 0.1 | 93.0 | 0.2 |
| Ours (RGB with 2 Render) | 97.7 | 96.4 | 1.3 | 97.5 | 0.2 |

Table 9: Performance comparison across seen and unseen scenarios with texture variations.

## F    Computational Overhead

During the action-related fine-tuning training stage, using LIBERO-Spatial as an example, the single S-RGB setting requires 8 A100 GPUs for approximately 20 hours during the video generation adaptation stage and an additional 12 hours for the action learning stage.

For video generation inference, the single-view setting consumes approximately 12 GB of GPU memory, while generating three views requires about 13.5 GB. The generation of a single video chunk takes around 20 seconds per view.

In action inference, the single S-RGB setting uses 10.6 GB of GPU memory, whereas the three-view configuration requires 12 GB. Action inference for a single S-RGB setting takes approximately 300 ms per action chunk, with a default chunk size of eight frames.

## G    Further Details on the Model Architecture

The main architecture is based on DynamiCrafter [53], with extensions to support multi-view processing using the Ray Camera Map and Spatial Attention. No additional specialized designs were introduced; instead, operations were conducted in a 4D latent space. Specifically, the input latent has a shape of `BCVTHW`, where B is the batch size, C is the channel, V is view number, and T stands for n timestamp. This input latent is reshaped as follows:

- $(BT)(VHW)C$ for spatial attention.

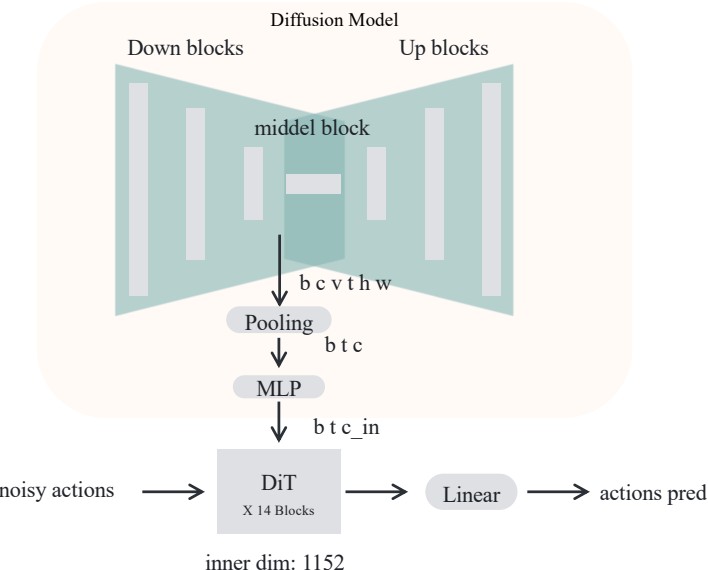

Figure 12: The Construction of the Action Policy Head.

| Hyperparameter | Configuration |
|---|---|
| Diffusion Setup | - Diffusion steps: 1000; Noise schedule: Linear; $\beta_0 = 0.00085$; $\beta_T = 0.0120$ |
| Sampling Parameters | - Sampler: DDIM; Steps: 500 |
| Input | - Video resolution: $320 \times 512$; Chunk size: 8; Encoded with VAE |
| | - Language prompt $c$, tokenized with T5 |
| | - Camera parameters encoded with ray direction map (L118 in main text) |
| UNet | - Latent image channels: 4; Ray map channels: 6; $z$-shape: $40 \times 64 \times 4$ |
| Temporal Attention | - Attention resolutions: $\{64, 32, 16\}$; Head channels: 64; Conv layers: 4 |
| | - Temporal kernel size: $3, 1, 1$; downscales: $40 \times 64 \rightarrow 20 \times 32 \rightarrow 10 \times 16$ |
| Spatial Attention | - Attention resolutions: $\{64, 32, 16\}$; Head channels: 64; Conv layers: 4 |
| Video Training | - Learning rate: $5 \times 10^{-5}$; Optimizer: Adam; Batch/GPU (single-view): 8; Batch/GPU (multi-view): 1 |
| | - Parameterization: $v$-prediction; Max steps: 100,000; Gradient clipping: 0.5 (norm) |
| Policy Training | - Same as video training, but with sample-prediction parameterization |
| Number of Parameters | - Base model (DynamiCrafter): 1.4B; Policy head (DiT blocks): 190M; VAE (frozen): 83.7M |

Table 10: Training details and hyperparameters used in our experiments.

- $(BVHW)TC$ for temporal attention.

- $(BVT)CHW$ before decoding.

The image decoding occurs per view and per frame. Features (BCVTHW) were extracted before the U-Net's middle block, followed by Pooling and an MLP to obtain a BTC' feature vector for conditioning the denoising process. The action head consists of 18 DiT blocks, with denoised latents passed through a linear layer for action predictions. As mentioned in Section 2.3, the header predicts the delta pose. Actions are represented as a 7-dimensional vector in pose space: delta position $(x, y, z)$, rotation (roll, pitch, yaw), and gripper openness. A simple diagram is shown in Fig. 12. We further provide the hyperparameters in Table 10.

## H    More Details on Training Data

**Pretraining Datasets.**    We pretrain on heterogeneous embodied datasets with clear task logic: RT-1 [4], Language Table [27], Bridge [46], RoboTurk [28], ManiSkill [14], and an Isaac Sim dataset [31]. Summary statistics:

- RT-1: 3.7M frames, 87K episodes, egocentric, real robot.
- Language Table: 7.0M frames, 442K episodes, front-facing, real robot.
- Bridge: 2.0M frames, 25K episodes, egocentric, real robot.
- RoboTurk: 72K frames, 1.9K episodes, front-facing, real robot.
- ManiSkill: 4.0M frames, 30K episodes, front-facing, simulation.
- Isaac Simulator: 3.0M frames, 40K episodes, egocentric + 8 third-person views, simulation.

At the time of this work, public embodied datasets predominantly provide single third-person or egocentric views, which are insufficient for training multi-view generation. LIBERO contains fewer than 500 trajectories, making it inadequate for robust multi-view learning. To bridge this gap, we constructed a multi-view dataset in Isaac Sim with ground-truth camera parameters. The simulator corpus contains about 40K episodes across 8 tasks spanning industrial and home scenarios, with diverse object layouts, lighting, and camera poses. Task list:

- place trash into the dustbin;
- pick fruit into basket;
- pick toy into box;
- insert pen;
- place bag;
- open drawer;
- place fruits;
- arrange workpieces.

We provide visual examples of data collected from the simulator in Fig. 13, with additional videos available in the supplementary material. At the time of this work, all available embodied datasets provided only single third-person camera views, which are insufficient for multi-view generation tasks. Furthermore, the evaluation benchmark LIBERO contains fewer than 500 trajectories, which is inadequate for training a multi-view generation model. Collecting real-world multi-view data directly is prohibitively expensive and labor-intensive.

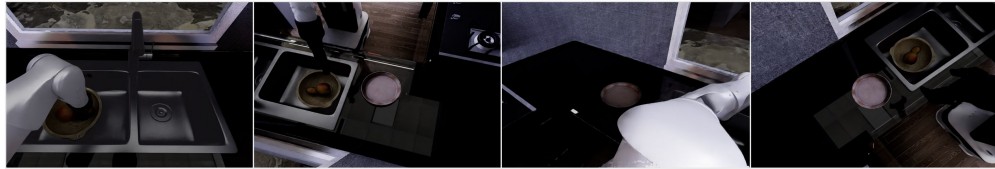

Figure 13: Visual Examples from the Simulator Collected Data.

## I    Visual Samples for our Data Engine

We provide visual samples from our Data Engine in Fig. 14. As shown in the figure, using ENER-VERSE-D as the data engine results in fewer artifacts and clearer boundaries.

Furthermore, we conducted additional experiments on the "arrange workpieces" task, where a robotic arm manipulates gears and boxes on a tabletop with frequent self-occlusions. Following the data-flywheel setting, given the task description and one complete head-camera video, the goal is to generate the corresponding video from a target view. We evaluated 50 generated episodes under two

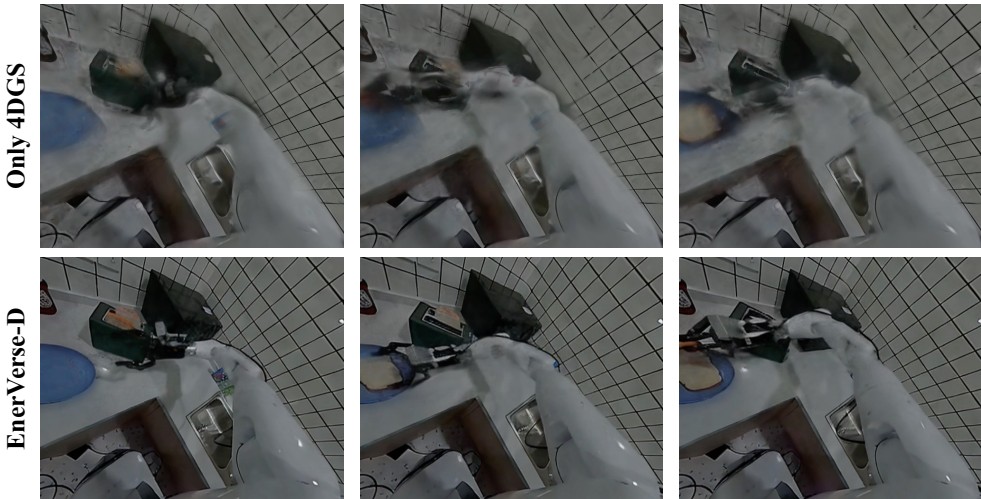

Figure 14: Visual Samples for Data Engine.

settings: (i) Without 4DGS—directly running the ENERVERSE-D video generation pipeline; (ii) With 4DGS—first generating an initial video via ENERVERSE-D, then applying the 4DGS pipeline to render target views, re-noising the renders, and feeding them back into ENERVERSE for refinement. Two blinded human experts assessed diffusion-induced hallucinations in the generated videos. The assessment result shows that integrating 4DGS reduces hallucinations by 40% relative to the baseline without 4DGS in scenarios with self-occlusions, quantitatively demonstrating the value of 4DGS in mitigating generative artifacts.

