# OpenReview forum: "EnerVerse: Envisioning Embodied Future Space for Robotics Manipulation"
_NeurIPS.cc/2025/Conference — NeurIPS 2025 poster_

### Official Review · Reviewer_y6Qo · 2025-06-26

**Clarity:** 3
**Significance:** 3
**Originality:** 2
**Rating:** 4
**Confidence:** 4

**Summary:**

The paper introduces EmerVerse, a generative robotics foundation model designed to construct and interpret embodied spaces for robotic manipulation tasks. The model uses a chunk-wise autoregressive video diffusion framework enhanced by a sparse context memory mechanism for long-term reasoning. It also integrates a multi-view video representation to address 3D grounding challenges and a 4D Gaussian Splatting (4DGS)-based data engine (EmerVerse-D) to bridge the sim-to-real gap. The policy head (EmerVerse-A) translates 4D world representations into physical actions, achieving good performance in simulation and real-world tasks.

**Questions:**

1. What is the core technical contribution beyond combining existing techniques?

2. How much does each component (sparse memory, multi-view, 4DGS) contribute to performance?

3. How sensitive is performance to different camera poses?

**Ethical Concerns:**

["NO or VERY MINOR ethics concerns only"]

**Final Justification:**

While some complexity remains and direct comparisons to very recent 3D video methods are limited, the overall strong empirical results and thorough rebuttal support a borderline accept.

**Limitations:**

Yes

**Quality:**

4

**Strengths And Weaknesses:**

**Strengths**

1. The chunk-wise autoregressive diffusion framework with sparse memory is a well-executed approach for long-term reasoning in robotic tasks. The multi-view diffusion generator and 4DGS-based data engine are well-motivated solutions to key challenges in robotics.

2. The paper provides extensive experiments across multiple benchmarks (LIBERO, CALVIN) and real-world tasks, demonstrating superior performance over existing baselines. Quantitative and qualitative results strengthens the claims.

3. The paper is well-written and clearly presents its methodology and results.

---

**Weaknesses**

1. **Technical Novelty**: While EmerVerse demonstrates impressive engineering efforts and strong empirical results, its technical novelty is limited as it primarily combines existing techniques—video diffusion, multi-view rendering, and 4D Gaussian Splatting. The work’s strength lies in its system-level integration and scaling, but the core methodology does not significantly advance beyond established paradigms. The authors are suggested to clarify the key technical novelty compared to existing works.

2. **Complexity & Computational Inefficiency**: While EmerVerse demonstrates strong empirical performance, its complexity and computational inefficiency raise concerns about practicality and novelty. The pipeline combines multiple heavyweight components—autoregressive video diffusion, multi-view rendering, 4D Gaussian Splatting, and a policy head—without clear justification for why this level of complexity is necessary. Inference is slow (20 seconds per multi-view chunk), making real-world deployment challenging. Crucially, the paper lacks ablation studies to show whether simpler, faster baselines could achieve comparable results, leaving the impression of an over-engineered solution rather than a fundamentally novel advance.

3. **Lack of Baseline Comparisons**: The paper notably omits comparisons with recent state-of-the-art 3D-aware video generation methods like Gen3C [1], which achieves precise camera control and world consistency through explicit 3D representations. A direct ablation against 3D-consistent generators would better contextualize its contributions.


[1] Ren, X., Shen, T., Huang, J., Ling, H., Lu, Y., Nimier-David, M., Müller, T., Keller, A., Fidler, S. and Gao, J., 2025. Gen3c: 3d-informed world-consistent video generation with precise camera control. In Proceedings of the Computer Vision and Pattern Recognition Conference (pp. 6121-6132).

---

> ### Author Rebuttal · Authors · 2025-07-31
>
> We greatly appreciate the reviewer’s recognition of our work and the constructive feedback provided. We have thoroughly reviewed your comments and address each point with detailed responses below.
>
> >Q1: While EmerVerse demonstrates impressive engineering efforts and strong empirical results, its technical novelty is limited as it primarily combines existing techniques—video diffusion, multi-view rendering, and 4D Gaussian Splatting.
>
> We thank the reviewer for acknowledging our engineering efforts and strong empirical results. However, our work goes beyond a simple combination of existing components. Our central contribution is the introduction of a video-generative robotic world model that leverages video prediction to model the robot’s future visual latent space. This model is pretrained on large-scale robotic data, enabling effective domain adaptation from general video generation to robotic world.
>
> The learned latent space forms the basis for policy learning. By attaching a lightweight, plug-and-play action head, we map the visual latent representation to actionable motor commands. This results in a unified framework that unifies robot sense and policy learning within a coherent pipeline, purposefully designed for real-world robotic manipulation. Each module is carefully introduced to support robotic world modeling, rather than assembled through ad-hoc integration.
>
>
>
>
> >Q2: While EmerVerse demonstrates strong empirical performance, its complexity and computational inefficiency raise concerns about practicality and novelty.
>
> We thank the reviewer for acknowledging that our framework could achieve strong empirical performance. However, we have to point out the 4DGS is not used for our data flywheel components which is generally an offline procedure. And during policy inference, the video generation process is not necessary. Instead, as stated in Line 144, we only use the feature vector after the first video denoising step and this feature vector is then cachaed and used as the condition feature vector for multiple action denosing steps. The action is predicted with the action chunks (chunk size is 8 in our setting, means we predict 8 step actions from one inference). We achieve 8-step action prediction within 280 ms on a real-world robotic platform equipped with an NVIDIA RTX 4090, which is comparable to the current mainstream VLA model, which is also generally about 200ms - 350ms （for example, OpenVLA[1] is about 180ms, DexVLA[2] is about 310ms）. Furthermore, we demostrate its effectivess with our real-world robotic manipulation tasks.
>
>
>
> >Q3: Lack of Baseline Comparisons: The paper notably omits comparisons with recent state-of-the-art 3D-aware video generation methods like Gen3C...
>
> We thank the reviewer for this suggestions. The concurrent work Gen3C [3] is open-sourced in GitHub on about Middle June, which is already after the submission deadline of the conference. Moreover, the goals of embodied video generation differ substantially from those of general video generation. Embodied settings place greater emphasis on scene consistency and task-relevant motion, which are not well captured by conventional video generation metrics. Due to the lack of multi-view baselines, we instead compared against SOTA single-view video generators: Kling-1.6, Hailuo I2V-01live, and Open-Sora-2.0. These models were conditioned on the same textual and visual prompts as our method to generate robot-centric videos. We then conducted a human preference study involving 20 annotators, who rated the outputs across dimensions such as scene consistency and object motion. Our method received 75 scores overall, outperforming Kling (64), Hailuo (30), and Open-Sora (1). Besides, in Gen3C, the camera moves around the target object, and the target object is generally with few movements, which differs greatly from our robotic manipulation tasks, where we use static cameras and robotic arm moves. We will add further discussions and comparsions in our revised versions.
>
>
> >Q4: How much does each component (sparse memory, multi-view, 4DGS) contribute to performance?
>
> Thank you for this question. We’ve carried out ablations to isolate the impact of each component:
>
> - Sparse Memory: As shown in Figure 7 (video rollouts) and Table 4 (policy learning), adding sparse memory substantially improves long-horizon generation quality and yields a significant boost in task success.
> - Multi-view: Table 2 compares single-view versus multi-view rollouts on LIBERO: incorporating multi views reduces raises success rates, especially on the LIBERO-10 splits.
> - 4DGS: We provided some inintal visualization in the Appendix, Figure 14. We also conducted some qualitative analysis， and finding that 4DGS would enhance the cross-view geometric consistency, and  dramatically reduces diffusion-induced hallucinations (especially under robotics arm - object occlusions). We will enrich the revised manuscript with additional visualizations of 4DGS’s effects and deeper analysis.
>
> >Q5: How sensitive is performance to different camera poses?
>
> We appreciate the reviewer's insightful question regarding the sensitivity of our method's performance to varying camera poses. To address this, we would like to highlight two key points that demonstrate the robustness of our approach against camera pose variations: Firstly, in the LIBERO benchmark, the front view camera mounted positions differ across different task scenarios. Furthermore, as noted in Lines 207 and 225, our policy evaluation leverages both the LIBERO and CALVIN benchmarks. Notably, the camera poses in these two benchmarks are fundamentally different.  The consistent performance of our method across these distinct setups directly validates its generalization capability to camera pose variations. Secondly, in our real-world experiments, we intentionally avoided labor-intensive camera calibration throughout the multi-day evaluation process. Instead, we relied solely on the default parameters provided by the robot's hardware client, which inherently contain non-negligible noise. The stable performance observed under such uncalibrated conditions further confirms the robustness of our method to practical camera pose inaccuracies.
>
>
> ## References
> [1] OpenVLA: An Open-Source Vision-Language-Action Model
>
> [2] DexVLA: Vision-Language Model with Plug-In Diffusion Expert for General Robot Control
>
> [3]  Gen3c: 3d-informed world-consistent video generation with precise camera control

---

> > ### Comment · Reviewer_y6Qo · 2025-08-06
> >
> > Overall, the rebuttal strengthens confidence in the paper’s technical validity, empirical thoroughness, and contribution significance. Given these clarifications and additional analyses, I support a borderline accept recommendation.

---

> > > ### Author Response · Authors · 2025-08-06
> > >
> > > Thank you for your positive feedback and for recognizing the strengthened technical validity, empirical thoroughness, and contribution significance of our work following the rebuttal. We truly appreciate your support and constructive suggestions, which have been invaluable in improving the paper.
> > >
> > > If there are any additional comments, questions, or points you would like us to address, please don’t hesitate to let us know. We are fully committed to further refining the work.
> > >
> > > Thank you once again for your time, effort, and thoughtful review.
> > >
> > > Best regards,

---

### Official Review · Reviewer_J1q3 · 2025-07-02

**Clarity:** 4
**Significance:** 4
**Originality:** 4
**Rating:** 6
**Confidence:** 5

**Summary:**

This paper proposes *EnerVerse*, a generative foundation model for robotic manipulation that predicts embodied future spaces via a chunk-wise autoregressive video diffusion framework augmented with sparse memory. It introduces Free Anchor Views (FAVs) for flexible multi-view 3D scene modeling and integrates a 4D Gaussian Splatting pipeline (EnerVerse-D) to reduce the sim-to-real gap. Coupled with a policy head (EnerVerse-A), the approach achieves state-of-the-art performance on manipulation benchmarks, demonstrating robust long-horizon reasoning and real-world deployment capabilities.

**Questions:**

1. Could the authors clarify why they claim that “wrist-mounted cameras complicate policy learning”? Are there any supporting studies or empirical evidence for this assertion?
2. Does multi-view generation induce severe hallucinations in long-sequence tasks, and if so, does this directly degrade action generation quality？

**Ethical Concerns:**

["NO or VERY MINOR ethics concerns only"]

**Final Justification:**

After reading the rebuttal and the other reviews, I am raising my score to 6. The paper provides convincing evidence that a world model can be trained and used for manipulation. The extensive real‑world experiments help mitigate the ambiguity common in simulation benchmarks (as noted by Reviewer C5W7). I also appreciate the use of multiple cameras to build the world model; this is a more realistic setup, whereas much prior work assumes a single camera. Overall, I recommend acceptance.

**Limitations:**

See weakness and questions.

**Paper Formatting Concerns:**

None.

**Quality:**

4

**Strengths And Weaknesses:**

**Strength:**

- The paper presents a comprehensive view from data generation to video synthesis and action planning, featuring a novel chunk-wise autoregressive diffusion model with sparse contextual memory for embodied future reasoning.
- The integration of 4DGS optimization into multi-view video generation establishes an effective data flywheel, leading to high-quality, realistic videos.
- The proposed FAV-based embodied future space approach, linked with policy planning, enhances spatial understanding and demonstrates clear potential for robotics applications.

**Weakness:**

While the proposed work is promising, further real-world robotic experiments are necessary. These experiments would bolster the evidence that the data generation pipeline effectively mitigates the sim-to-real gap, which is one of the primary motivations for this research.

---

> ### Author Rebuttal · Authors · 2025-07-31
>
> We sincerely thank the reviewer for recognizing the contributions of our work and for providing constructive comments. We have carefully considered your suggestions and provide detailed, point-by-point responses below.
>
> >Q1: further real-world robotic experiments are necessary.
>
> We thank the reviewer for this valuable suggestions. In the current submission version, we conducted real-world experiments on the tasks Block Placing, Plastic Objects Sorting, and Fruit Sorting as shown in *Section A in the Appendix*. And we are trying to add new real-world experiments including *Bin Picking*， *Package Sorting*, etc. The visual results will be updated in our revised version paper.
>
> >Q2: Clarify why they claim that “wrist-mounted cameras complicate policy learning”?
>
> We thank the reviewer for this questions. This statement is coming mainly from our setting, where the input for the model is the text prompt c, observation images o and the related inaccurate camera parameters. Those camera paramters would serve as the condition information for the multi-view generation diffusion process, then the denoised feature vector will guide the final cation planning, e.g. the pose of the EEF.  If we feed the images from the wrist camera, we also need to input its camera parameters. However, its extrinsic parts has directly replected the EEF's pose, which is slightly violating our settings. Besides, in our practical training experiments, we found that when we use both third-view images and wrist images for the training process, we would result in degraded generation performance, and the final policy performance is also suboptimal. As a result, our framework is more suitable to receive multi static view image input, similar to [1]
>
> >Q3: Does multi-view generation induce severe hallucinations in long-sequence tasks, and if so, does this directly degrade action generation quality？
>
> We appreciate the reviewer’s concern. In fact, our multi‐view generator is built with cross‐view attention and trained on synchronized multi‐view video data, which strongly enforces geometric constraints and thus reduces hallucinations consistent with prior work on multi‐view consistency[1,2]. As shown in Figure 6, our model achieves high consistency across views, confirming its cross‐view coherence even over long sequences. Crucially, this consistency translates into better action generation: on the LIBERO-Long splits, a policy using only single-view S-RGB rollouts achieves 73 % success, whereas augmenting with two additional rendered views boosts success to 80 %. The improvement is largest compared with other splits, demonstrating that our multi-view conditioning enhances (not degrades) downstream control performance by mitigating hallucinated dynamics. Furthermore, in practice, unlike purely video generation setup,
>
> ## References
> [1] RVT: Robotic View Transformer for 3D Object Manipulation
>
> [2] Zero-1-to-3: Zero-shot One Image to 3D Object
>
> [3]GeoWizard: Unleashing the Diffusion Priors for 3D Geometry Estimation from a Single Image

---

### Official Review · Reviewer_hqTx · 2025-07-02

**Clarity:** 1
**Significance:** 4
**Originality:** 4
**Rating:** 4
**Confidence:** 2

**Summary:**

This submission focuses on learning a multi-view latent space for robotic manipulation using a world modeling objective. The paper proposes three systems: EnerVerse, a diffusion world model that takes patches from prior images and predicts future patches based on a context variable, EnerVerse-A, a diffusion transformer policy trained on top of the EnerVerse latent space, and EnerVerse-D, a system for using Gaussian Splatting to generate novel views used to train the world model. Overall, this submission describes a large system. EnerVerse is tested in its ability to predict future frames and in terms of being able to learn generalizable policies. Experiments are performed in LIBERO and CALVIN in simulation as well as on a real-world AGIBOT platform.

**Questions:**

Could you add substantially more details to your description of EnerVerse world model and EnerVerse-D?

What is the conditioning variable $c$? Is it possible to condition the world model on actions?

Is it possible to use the world model to generate synthetic training data for the policy?

**Ethical Concerns:**

["NO or VERY MINOR ethics concerns only"]

**Final Justification:**

I think this submission contains novel methodological contribution as well as significant and interesting results. The main drawback is that the submission describes a large and complicated data generation and robot learning system, and the authors provide limited details about (a) the simulated dataset used to train the model and (b) the "data flywheels" used to generate additional data. This was also pointed out by Reviewer C5W7. The authors provided more details in the rebuttal but the discussion is still at a fairly high level. Therefore, the community may benefit from some ideas in the paper, but it is highly unlikely that the system described in this paper will be replicated in future works.

**Limitations:**

The authors describe inaccuracies of video prediction methods and heuristic camera pose selection as the main limitations of their work in Section 5.

**Paper Formatting Concerns:**

None.

**Quality:**

3

**Strengths And Weaknesses:**

## Strengths

**Novel sparse memory mechanism.** It is proposed to subsample patches from each frame in the video history so that a longer history can be provided to the model. This idea is helpful both because it allows for a longer history and also because it serves as a data augmentation method (by randomizing which patch is selected) leading to better generalization. This design choice is ablated in Table 4, showing that having longer memory leads to a major improvement in policy learning success rate. Further, Figure 7 shows a convincing example of the world model remembering that a stove was turned on at a prior time step.

**Strong multi-view video generation results.** Figure 6 depicts an example of multi-view video generation that demonstrates object consistency across multiple views in a simulated and a real-world scene. This is supported by a quantitative evaluation in Table 1.

**Strong policy learning results.** The authors demonstrate that policies trained based on the latent space of EnerVerse achieve state-of-the-art success rates on the LIBERO benchmark. LIBERO specifically tests generalization to different objects, layouts, goals and background and this is a strong indication that EnerVerse learns generalizable features. Further, CALVIN results in Table 3 are on par with strong baselines.

## Weaknesses

**The method description is incomplete.**
* EnerVerse: It is stated that EnerVerse is based on DynamiCrafter, but the paper does not actually specify the size of the EnerVerse model (in the number of parameters) or any training details (training steps, optimizer setting, dataset size, etc.). The only training details provided are about the EnerVerse-A policy head. Further, I believe the paper does not specify what is the content of the context variable $c$ that is used to condition the video prediction model.
* EnerVerse-D: Section 2.2 describes the high-level idea of generating multiple views using a video prediction model, training a 4D Gaussian splat on this sequence and generating novel views from the Gaussian splat to further train the video prediction model. However, the submission does not contain specific details as to how this system was implemented. This limits the impact of this paper, as it is unlikely that researchers could reproduce the system from its description.

**The experiments description is incomplete.** On line 157, it is stated “Furthermore, we constructed a dataset containing multi-view video ground truths using the Isaac Sim simulator [31].” The submission does not contain further elaboration as to the size of the simulated dataset or the diversity of the generated scenes. Therefore, it is unlikely that researchers could independently reproduce this training setup.

## Minor comments:
* Line 86: define $J$.
* Line 93: define $v$.

---

> ### Author Rebuttal · Authors · 2025-07-31
>
> We appreciate the reviewer's thoughtful feedback. We have thoroughly addressed the concerns regarding methodological clarity and experimental coverage in the responses below, and note that some supporting details are already included in the appendix. We will revise the main paper to improve clarity and ensure all aspects are clearly presented.
>
>  >Q1: More Training details
>
> We thank the reviewer for this valuable question. And we provide the training details and hyperparameters in the following table, and this table will be added to the main paper in the revised version.
>
> | Hyperparameter       | Configuration                                                                                                 |
> |----------------------|---------------------------------------------------------------------------------------------------------------|
> | **Diffusion Setup**  | - Diffusion steps: 1000  - Noise schedule: Linear  - beta_0: 0.00085  - beta_T: 0.0120 |
> | **Sampling Parameters** | - Sampler: DDIM  - Steps: 500                                                           |
> | **Input**            | - Video, resolution: 320 x 512  - Chunk size: 8  - Encoded with VAE |
> |                      |- Language prompt c, tokenized with T5  |
> |                      | - Camera Parameters, encoded with ray direction map (L118 in main text) |
> | **UNet**             |  - Latent image channels: 4  - Ray map channels: 6  - z-shape: 40 x 64 x 4 |
> | **Temporal Attention** | - Attention resolutions: 64, 32, 16  - Head channels: 64  - Conv layer num: 4  - Temporal kernel size: 3,1,1  -> 40 x 64 -> 20x32 -> 10x16 |
> | **Spatial Attention** | - Attention resolutions: 64, 32, 16  - Head channels: 64  - Conv layer num: 4 |
> | **Video Training**       | - Learning Rate: 5x 10e-5  ; - Optimizer: Adam ; - Batch size per GPU (single-view): 8 ; - Batch size per GPU (multi-view): 1 |
> |                      |  - Parameterization: v-prediction  ; - Max steps: 100,000 ; - Gradient clipping: 0.5 (norm)|
> | **Policy Training** | Similar to video training, but with sample-prediction parameterization|
> | **Num of Parameters** | - Base model: DynamiCrafter 1.4B  - Policy Head: 190M (DIT Blocks)   - VAE: 83.7M(frozen)|
>
> >Q2: Details on the dataset
>
> We sincerely thank the reviewer for this question. And we provide the pre-training dataset details as follows. And we will add these information in our main text in the revised version.
>
>
> | Dataset         | Frames   | Episodes   | Camera View                     | Category    |
> |-----------------|-------------------|------------|---------------------------------|-------------|
> | RT-1            | 3.7M              | 87K        | Egocentric                      | Real robot  |
> | Language Table  | 7.0M              | 442K       | Front-facing                    | Real robot  |
> | Bridge          | 2.0M              | 25K        | Egocentric                      | Real robot  |
> | RobotTurk       | 72k               | 1.9k       | Front-facing                    | Real robot  |
> | ManiSkill       | 4.0M              | 30K        | Front-facing                    | Simulation  |
> | Issac Simulator | 3M                | 40k        | Egocentric + 8 Third Views     | Simulation  |
>
> >Q3: context variable c
>
> In Line 89, we descibe the c as the textual prompt. In our implementation, this textual prompt is encoded by a frozen T5-base and an MLP is used to project the output to the token space.
>
> >Q4: Details on EnerVerse-D implementation.
>
> After pretrain the EnerVerse-G, it is capable to receive m-view images (with the camera paramters), task prompt and then produce the predicted m-view videos. Following the standard diffusion process, we add noise to the latent from all m-view images. However, in the data flywheel setting, we have at-least one (N) view camera mounted on the robot has complete and clean visual observation images, and we did not need to predict the future frames for this view and thus *NO* noise is appied on the correponding latents. Instead, those visual observation will serve as a guidance for other views video generation. We finetune our base model to let it capable to receive such input format. In view of 4D latent space with dimension of CVTHW, for tensor coming from the camera images input, e.g. C1THW, we did not apply noise.
>
> During inference, we use clean N-view camera images and their camera extrinsics to obtail M-view rendered images with the depth wraping, as shown in Figure 4. These M+N view videos with text prompt c then go through the diffusion model to obtain the cleaner denoised M-view images. After that, these M+N view videos and their cooresponing camera parameters are used to construct the 4DGS representation. We use the officaial implementation of 4DGaussian [1], and followed most default settings, the only change is that we use the depth information to do the initialization process and the depth of deformation model is set to 1. The 4DGS model produces higher-precision, geometrically consistent 4D representations. Then, we again render M-view images from the 4DGS to obtain higher-quality observations. These M+N video videos then agian go through diffusion process and 4DGS process to iteratively improve the quality.
>
> >Q4:  The submission does not contain further elaboration as to the size of the simulated dataset or the diversity of the generated scenes.
>
> We thank the reviewer for this valuable suggestions. The reason why constructed that simulated dataset is that, At the time of this work, all available embodied datasets provided only single third-person camera views, which are insufficient for multi-view generation tasks. We provided visual examples for the simulator collected data in Appendix Figure 13, and this dataset is with about 40K episodes. The collected episodes is with 8 tasks, and crossing industrial and home scenarios. The task descriptions are as following, and we plan to release some splits in the near future to enable other researchers could independently reproduce this training setup.
>
> - Task Descriptions:
>     1. place trash into dustbin
>     2. pick fruit into basket
>     3. pick toy into box
>     4. insert pen
>     5. place bag
>     6. open drawer
>     7. place fruits
>     8. arrange workpieces
>
> >Q5: Symbol defination in Line 86  and Line 93.
>
> The **J** in Line86 describes the frame number of rendered images. We used the **J** for rendered views $r_t^{1:J}$ instead of **K** used for $o_t^{1:K}$ is that the rendered views and the obseravtion camera images could have different frames. For an example, in the EnerVerse-D, the camera obervations have complete offline videos, in such case $K=T$, while the rendered views could only have one frame.
>
> We follow [2] to implement the v-prediction. Instead of predicting the noise $\epsilon_t$, the model predicts $v_t$, defined as: $v_t = \alpha_t \epsilon_t - \sigma_t x_0$.
>
> >Q6:  Is it possible to condition the world model on actions?
>
> We thank the reviewer for this insightful question. While our current framework focuses on generating videos and subsequently predicting action policies from text instructions and image inputs, we believe it is easy to adapt to support action-conditioned video generation. Although this is not the primary setting explored in our work, the extension is conceptually straightforward: if physical action sequences can be represented in text or visual modalities, they can be directly integrated into our framework, enabling an action-to-video generation process.
>
> Moreover, we view our framework's ability to ground text prompts and visual observations as a foundational capability for building action-conditioned video models. We consider this a promising direction [3] and plan to explore it in future work.
>
>
> >Q7: Is it possible to use the world model to generate synthetic training data for the policy?
>
> Thank the reviewer for this valuable questions. It is a promising and active research domain to use world model to generate synthetic training data for the policy learning as done ine [4], where the model generates both video frames and action lables. For this work, we chose to focus more on the action output, e.g. the robot policy learning.
>
> ## References
>
> [1] 4D Gaussian Splatting for Real-Time Dynamic Scene Rendering
>
> [2] Progressive distillation for fast sampling of diffusion models
>
> [3] 1X World Model: Evaluating Bits, not Atoms
>
> [4] DreamGen: Unlocking Generalization in Robot Learning through Video World Models

---

> > ### Comment · Reviewer_hqTx · 2025-08-05
> > **Response**
> >
> > Thank you for the rebuttal, the additional information is helpful. Q1, Q2 and Q4 are very important points that should be included in the paper, since the significance of the original submission was limited by these missing details.

---

> > > ### Author Response · Authors · 2025-08-05
> > >
> > > Thank you for your thoughtful feedback and for acknowledging the additional information provided in the rebuttal. We greatly appreciate your suggestions regarding Q1, Q2, and Q4. We will make sure to incorporate these important points into the revised manuscript to address the limitations of the original submission.
> > >
> > > If you have any further questions or additional feedback, please don’t hesitate to let us know. We would be more than happy to provide further clarification or address any remaining concerns.

---

> > > > ### Comment · Reviewer_hqTx · 2025-08-06
> > > > **Response**
> > > >
> > > > I increased my score and I am in favor of accepting this paper.

---

### Official Review · Reviewer_C5W7 · 2025-07-04

**Clarity:** 2
**Significance:** 3
**Originality:** 3
**Rating:** 5
**Confidence:** 3

**Summary:**

This paper presents EnerVerse, a robotics foundation model that is capable of generating future frames conditioned on past observations + task context, and using these future frames for action prediction to solve the task. In particular, EnerVerse uses a chunk-wise autoregressive diffusion model with sparse memory for future prediction and incorporates multi-view video generation for better 3D spatial understanding. This video generation model can be extended to action generation (EnerVerse-A) by adding a diffusion-based action head. To facilitate better training of the multi-view frame generation + action prediction, the paper proposes a data synthesis framework for multi-view video generator based on sparse observations in simulation. Experiments are performed for video generation quality and robotic task execution, and show promising improvements over prior work.

**Questions:**

My initial assessment is that this paper is incomplete and not yet ready to be accepted. My primary concerns are to do with the paper clarity and experimental section. I have listed the questions along with the weaknesses.

**Ethical Concerns:**

["NO or VERY MINOR ethics concerns only"]

**Final Justification:**

I thank the authors for their patience and efforts in this process. After multiple rounds of discussion with the authors, several points were clarified about the methodological details. Authors also pointed to existing experiments from appendix to address some concerns (e.g., quantitative real-world results of OpenVLA vs. EnerVerse) and performed new experiments to address weaknesses in the experimental section (listed below).
* Standard deviations were included for OpenVLA and EnerVerse on LIBERO
* An additional video generative method (LTX) was included in addition to DynamiCrafter. The proposed EnerVerse method improves on the video generation quality with both LTX and DynamiCrafter. This further strengthens the significance of EnerVerse.
* 3D video generation (DynamiCrafter) was compared against 4D scene generation (EnerVerse) on robotics tasks (LIBERO-Spatial), and the value of 4D scene generation was clearly demonstrated for robotics control.
* To address the lack of common baselines between LIBERO and CALVIN, OpenVLA was evaluated on both benchmarks. EnerVerse outperforms in both cases.
* Impact of 4DGS was quantitatively studied through a very small-sample human study of hallucinations occurring with and without 4DGS. The authors have further promised to extend this analysis to impact for other experiments in the paper.

While I still have concerns about the final paper requiring significant changes to address the clarity concerns and add new experiments, I feel more confident to take the leap of faith that these will be incorporated into the paper effectively. Therefore, I am raising my rating to **Accept**.

**A request to the authors for the final paper version:** Since this was a short rebuttal turnaround period, I expect not all new experiments may have been run to completion / may have been smaller scale than needed. I hope the authors can have a more complete version of these experiments for the final paper. Some examples below:

* **LTX experiments:** *However, due to time and computational constraints during the first-round discussion period, we were unable to complete the full training of all variants. In this round discussion, we managed to provide the complete results. However, both LTX variants were trained for only ~30k steps, which is fewer than the fully trained DynamiCrafter setup due to the time limitations*

* **Human study for 4DGS impact:** *For evaluation, we asked two human experts to assess the generated videos specifically for diffusion-induced hallucinations.* --- it would be nice to have a larger sample size of human experts

* **Other quantitative studies for 4DGS:** *We acknowledge the need for more comprehensive quantitative analyses. To further support the benefits of 4DGS, we plan to conduct additional experiments on other tasks and metrics in the revised version of our paper.*

**Limitations:**

Yes

**Quality:**

3

**Strengths And Weaknesses:**

# Strengths
## Strength 1
 I liked some of the experimental aspects of the work.
* Real-world qualitative results are useful to understand the sim2real generalizability of the work.
* Several relevant robotic action policies are compared against in Tables 2 and 3, strengthening the claims of the paper and demonstrating the usefulness of high-quality multi-view video generation for robotic tasks.
* Additional experiments are presented to study the value of different components of the system, which are helpful to understand their impact on the robotic tasks (sparse memory in Table 4, training strategy in Table 5, and utility of multi-view rendering in Table 2).

## Strength 2
The related work section was good and helpful to understand the relevant literature on video generation and robotics with foundation models.

## Strength 3
I appreciate that the authors have included qualitative videos of robotic task executions in the supplementary material.

# Weaknesses

## Weakness 1:  Paper writing clarity is poor
The proposed approach was quite difficult to understand given the amount of partial / missing information in the main paper. This makes it hard for a reader to grasp the idea and the core contributions.
* **Q1:** L93 - The velocity term used for prediction is not described (in fact, the variable $v$ is used without any explanation or definition)
* **Q2:** Section 2.2 4D Model - Lots of things are unclear here.
	* What is the exact architecture of the "Multi-View Video Generator" in Figure 3? Some details are mentioned in text (like "spatial attention is applied ..." in L121-122), but these are very vague.
	* How exactly is the 4D model trained?
	* What are the inputs, outputs and losses during training?
	* What are the sources of training data for 4D training?
	* How are the multi-view camera extrinsics decided during training and inference?
* **Q3:** Section 2.2 Real-World Data Flywheels - The mechanism and motivation behind the data generator is vague.
	* L127 - "utilizes sparse observations ..." - where are these sparse observations coming from? How are these sparse observations being utilized?
	* L128 - What is the base model EnerVerse here? Is it the model trained for generating a single video from past video + task context?
	* L129 - How is EnerVerse "fine-tuned to be capable of receiving a complete offline observation sequence"?
	* L129 - 131 - "When inferring, .... used to construct a 4D ..." -- What does inference mean here? What are the inputs and outputs? How are they used to construct a 4D gaussian representation?
* **Q4:** Section 2.3
	* How is the model trained for action prediction?
	* What are the loss functions?
	* Does the diffusion happen along with the video diffusion or are they separate?
	* Does the action diffusion model use past actions as inputs?
* **Q5:** L183 - Unclear how the DynamiCrafter baseline works in this context.
	* DynamiCrafter is trained to accept an image and text context to generate short videos. How exactly is it adapted as a baseline to this setting with long video generation?
	* Does DynamicCrafter baseline get the advantage of the same training data as EnerVerse (see L154 - 156)?

## Weakness 2: Experiment section feels incomplete
While I liked some aspects of the experiment design, there are a lot of things missing / unclear.
* **Q1:** Why is DynamiCrafter the only video generation baseline? Why are other models not evaluated? AVID (https://arxiv.org/html/2410.12822v2), I2VGen-XL (https://arxiv.org/abs/2311.04145), Stable Video Diffusion (https://arxiv.org/abs/2311.15127), Motion-I2V (https://dl.acm.org/doi/abs/10.1145/3641519.3657497), LivePhoto (https://link.springer.com/chapter/10.1007/978-3-031-72649-1_27), PhysGen (https://link.springer.com/chapter/10.1007/978-3-031-73007-8_21), etc. to name a few.
* **Q2:** Multi-View Generation is stated as a key contribution (L74), but there is only a small qualitative analysis in Figure 6 to evaluate this. Why are there no quantitative evaluations and comparisons to other baselines for novel view synthesis?
* **Q3:** One of the core assumptions here is that better video generation should result in better robotic task performance (L23 - 25). This has been completely neglected in the experiments. Ideally, multiple video generation baselines must have been evaluated in Table 1, and their corresponding performance on Tables 2 and 3 must be reported.
* **Q4**: Tables 2 and 3 do not report standard deviations in performance over multiple trials --- how significant are the differences in performances between methods? Are they within noise margins?
* **Q5:** The results in Tables 2 and 3 are a bit confusing because different rows use different visual inputs. It is hard to get an apples-to-apples understanding of the relative performances between methods. Is it possible to equip other methods with the depth sensor (i.e., S-RGBD)? Is it possible to equip EnerVerse with G-RGB?
* **Q6:** Why are baselines across Tables 2 and 3 different?
* **Q7:** L277 - Real-world experiments are only qualitative. Can we get quantitative results and comparisons to a subset of baselines in the real-world?
* **Q8:** L312 - What is the impact of the gaussian-splatting based approach to reduce the Sim2Real gap? This has not been studied in the experiments.
* **Q9:** There are no baselines that leverage multi-view representations. The idea of using multiple views to enhance model representations and robotic task performance is not new (e.g., MVWM: https://arxiv.org/pdf/2302.02408, RoboHorizon: https://arxiv.org/abs/2501.06605). How do these compare to the proposed approach?

---

> ### Author Rebuttal · Authors · 2025-07-31
>
> We thank the reviewer for their valuable comments. We have addressed the concerns on methodological clarity and experimental completeness in detail below, with some additional information already included in the appendix. We will revise and polish the main paper to ensure clarity.
>
> > Q1: velocity term
>
> As mentioned in L93, we follow [1] to implement the v-prediction. Instead of predicting the noise $\epsilon_t$, the model predicts $v_t$, defined as:
>
> $v_t = \alpha_t \epsilon_t - \sigma_t x_0$
>
> Here, $\alpha_t = \sqrt{\bar{\alpha}_t}$ (signal scale) and $\sigma_t = \sqrt{1 - \alpha_t^2}$ (noise scale), consistent with the forward process equation $x_t = \alpha_t x_0 + \sigma_t \epsilon_t$.
>
> > Q2: 4D Model Details
>
>  Due to space limitations, we moved the model architecture details to **Appendix L591–L604**. Additional details are as follows:
>
> * **Input and Output:** The model takes as input a task prompt c, $m$-view observation images($m \ge 1$)$O_t$, and their camera parameters, and outputs the predicted $m$-view videos.
> * **Loss:** We use the v-prediction diffusion loss to supervise $m$-view image generation, as noted in Q1.
> * **Dataset:** As stated in L155–L158, we use both single-view academic datasets and self-constructed multi-view datasets from the simulator. For single-view videos, camera extrinsics (relative to the robot base) are manually and approximately estimated; for simulator videos (examples in L606), GT parameters are available. During inference, we use the extrinsics provided by the robot hardware. We plan to release the simulator rendering scripts and estimated extrinsics.
>
> > Q3: Mechanism on data flywheels.
>
> After pretraining the 4D base model (e.g., EnerVerse-G) on the dataset mentioned in Q2, the model takes as input multi-view images (with camera parameters) and a task prompt, and outputs predicted multi-view videos. Following the standard diffusion process, we add noise to the latent features from all views. However, in the **data flywheel**, at least one camera (n ≪ m) mounted on the robot provides full visual observations. For these views, we skip noise injection and instead use their latents as guidance for generating other views. We refer to these as **sparse observations**.
>
> To support this input format, we fine-tune the base model. In the 4D latent space of shape CVTHW, no noise is applied to latents of the n observed views (e.g., C₁THW). During inference (see L153–Fig.4), for examlpe, we use one full video, two rendered initial frames, three camera parameters, and a task prompt. The model then generates denoised videos for the two target views. These, along with the original observed video and parameters, are fed into 4DGS to construct the 4D Gaussian representation. Then rendered frames are obtained from GS, then iteratively.
>
> > Q4: Policy Model Trianing.
>
> After obtaining EV-G (L139), we attach a policy head $h\_\theta$, composed of stacked DiT blocks, to enable action generation, resulting in EnerVerse-A. The goal of EnerVerse-A ($f\_\theta$) is to estimate the clean action from a noisy one:
>
> $
> a_t^0 \leftarrow f_\theta(c, O_t, a_t^k, k) = h_\theta(E, a_t^k, k),
> $
>
> where the latent vector \$E\$ is extracted fromthe U-Net backbone in EnerVerse-G. We train \$f\_\theta\$ by minimizing the denoising mean-squared error:
>
> $$
> \mathcal{L}(\theta) := \text{MSE} \left( a_t, f_\theta(c, O_t, \sqrt{\bar{\alpha}^k} a_t + \sqrt{1 - \bar{\alpha}^k} \epsilon, k) \right), \tag{2}
> $$
>
> where \$k \sim \text{Uniform}({1, \ldots, K})\$. As described in L146, we predict a chunk of actions in one shot to improve temporal consistency and reduce error accumulation by decreasing decision frequency.
>
> During inference, \$f\_\theta\$ takes $O\_t\$ and \$c\$, which pass through the video diffusion backbone once to compute the latent vector \$E\$. This \$E\$ is cached and reused across multiple denoising steps for action generation, and no past actions are used.
>
> > Q4:  DC for long-video generation.
>
> In L183, we primarily adopt FreeNoise [3], a tuning-free paradigm for extending video length via noise rescheduling. Its core mechanism involves local noise shuffling, constructing long-range correlated noise sequences, and applying window-based attention fusion. In our implementation, the second inference stage takes the results from the first stage as input, and FreeNoise is applied at each inference stage.
>
> > Q5: Does DynamicCrafter get the advantage of the same training data as EnerVerse?
>
> For a fair comparison, the EnerVerse reported in Tab 1 and Fig 5 is trained solely on RT-1. Based on the generation results, we attribute the performance gains primarily to our model’s architectural design.
>
> > Q6:  DC the only video baseline
>
> Our work targets world models for embodied agents on single-GPU or edge-GPU platforms, where video generation is not the core focus. While recent video diffusion models (e.g., I2VGen-XL, Stable Video Diffusion, LivePhoto, PhysGen) achieve high visual quality, they are often unsuitable for our setting—being too large (I2VGen-XL), reliant on extra components like simulators (PhysGen), or closed-source (LivePhoto). In contrast, **DynamiCrafter** is a well-maintained open-source model with public weights and serves as the basis for AVID. Considering our compute and latency constraints, we selected DynamiCrafter for reproducibility.
>
> Similar to prior works such as VPP[4] and VidMan [5], which adopt a single video model due to high training cost, we additionally selected **LTX** [6] for further analysis because of its efficiency and favorable inference speed for robotic tasks. We report PSNR results showing **EnerVerse-LTX (27.8) > EnerVerse-DC (26)**, likely benefiting from LTX’s advanced architecture and higher-quality pretraining, and demonstrating the flexibility of our method to different video models. These results and the discussion of additional models will be included in the revised paper.
>
> > Q7: No other baselines...(Multi-view)
>
> The availability of open-source multi-view video generation models with publicly released weights is limited. Most SOTA methods are either closed-source, require substantial computational resources for fine-tuning, or depend on bespoke rendering pipelines, making them impractical as baselines under realistic constraints.
>
> Furthermore, the objectives of embodied video generation differ significantly from general video synthesis. Embodied scenarios demand high scene consistency and task-relevant motion—factors not well captured by conventional evaluation metrics. Given the lack of suitable multi-view baselines, we compare against SOTA single-view generators: Kling-1.6, Hailuo, and Open-Sora, under identical textual and visual conditions. In a human preference study with 20 annotators, EnerVerse received 75 votes, outperforming Kling (64), Hailuo (30), and Open-Sora (1).
>
> While EnerVerse may be less competitive in general-purpose video generation, it is more aligned with the requirements of embodied video synthesis. Our primary objective is to validate the utility of multi-view prior synthesis for robotic policy learning. Accordingly, we present qualitative results to establish feasibility and leave more comprehensive quantitative evaluations—including those involving real-robot policy improvements—for future work as more suitable baselines become available.
>
> >Q8:  assumption better video ... better robotic task performance (L23)
>
> To clarify, in L23 we do not assume a naïve one-to-one mapping whereby any uplift in general video-generation metrics (e.g. FVD, SSIM) will automatically yield better robotic control. Rather, our claim is that representational alignment—i.e., adapting the video model’s latent space so that it encodes 3D, action-conditioned dynamics brings gains.
>
> > Q9: not report standard deviations
>
> The selected baselines, such as GR-1 and SUSIE, did not report standard deviations. To ensure a consistent comparison, we also omitted them in the main paper. Here, we provide results comparable to OpenVLA.
>
> | | Spatial | Object | Goal | Long |
> |-----------|-----------|--------|-|----|
> | OpenVLA | 84.7 ± 0.9    | 88.4 ± 0.8   | 79.2 ± 1.0 | 53.7 ± 1.3 |
> |Our | 92.1 ± 0.7 | 93.2 ± 0.9 |  78.1 ± 1.1 | 73.0 ± 1.2 |
>
> >Q10: use different visual inputs in Tab 2/3
>
> Our method with a single S-RGB input outperforms all other approaches, including those using S-RGB or S-RGB + G-RGB, demonstrating its effectiveness. Current most visual-policy or VLA models do not support depth input, and our current pipeline requires a static camera for G-RGB.
>
> >Q11: baselines across Tab 2 and 3 are different.
>
> In policy learning, CALVIN and LIBERO are two representative large-scale benchmarks, and most prior works evaluate on only one of them. To better demonstrate the effectiveness of our method, we conduct experiments on both.
>
> >Q12: quantitative results and comparisons to a subset of baselines in the real-world
>
> In Tab 6(L531), we provided the quantitative results for ours and OpenVLA.
>
> >Q13: The impact of GS
>
> We provide the visualization results in the Appendix. The main goal of GS is to enhance the geometric consistency and reduce video diffusion hallucinations (especially under robotics arm - object occlusions). We will provide more visual examples and deeper analysis in the revised version.
>
> >Q14: No baselines that leverage multi-view.
>
> In Tab 2, we provide the MAIL with two S-RGB results, and many baselines with S-RGB+G-RGB. MVWM focuses more on the masked autoencoder, while RoboHorizon emphasizes more on the LLM's usage.
>
> [1]Progressive distillation for fast sampling of diffusion models
>
> [2]Learning fine-grained bimanual manipulation with low-cost hardware
>
> [3]Freenoise: Tuning-free longer video diffusion via noise rescheduling
>
> [4] Video Prediction Policy: A Generalist Robot Policy with Predictive Visual Representations
>
> [5] VidMan: Exploiting Implicit Dynamics from Video Diffusion Model for Effective Robot Manipulation
>
> [6] LTX-Video: Realtime Video Latent Diffusion

---

> > ### Author Response · Authors · 2025-08-06
> > **Follow-up on Discussion**
> >
> > Dear Reviewer,
> >
> > I hope this message finds you well. As the discussion period is approaching its end with less than three days remaining, I wanted to kindly follow up to ensure we have addressed all your concerns comprehensively. If there are any additional points or feedback you'd like us to consider, please let us know at your earliest convenience.
> >
> > We greatly value your insights and are committed to addressing any remaining issues to improve our work. Thank you again for your time and effort in reviewing our paper.
> >
> > Best regards,

---

> > ### Comment · Reviewer_C5W7 · 2025-08-07
> > **Reviewer response to rebuttal**
> >
> > I thank the authors for their clarifications and additional experiments. The methodological details were largely clarified. Experimental concerns were partially clarified, but quite a few remain. I have highlighted my remaining concerns below. Aside from the concerns below, I'm also wondering what the paper would look like after all these clarifications are incorporated, i.e., there could be significant changes beyond minor clarifications and experimental additions (this has been noted by reviewer hqTx as well). I feel that it is not ready to be accepted yet.
> >
> > * **4D model details:** Thanks for the clarifications. The architecture explanation in Appendix L591 - L604 is also high-level and not thorough, and requires readers to have a perfect understanding of DynamiCrafter. I would expect a more thorough from-the-scratch explanation (with figures) of how the architecture works and what modifications were made compared to DynamiCrafter (and why). I would appreciate it if the authors can update this in a future version of the paper. I do not expect additional responses regarding this comment. I just wanted to share this concern.
> > * **DynamiCrafter the only video baseline:**
> > 	* Thanks for the clarification about latency requirements. Please add this to the paper.
> > 	* What is EnerVerse-LTX? Why is there no comparison to pure LTX without any of the contributions from this paper?
> > * **Better video -> better robotic task performance?** If this is not the assumption, then what are the alternative ways to evaluate video generation models in terms of their fit for robotics control? Also, if the video quality does not matter, why not compare DC-FN and LTX on the robotic control tasks directly?
> > * **CALVIN vs. LIBERO mismatch in baselines:** If the code bases for the baselines are open-sourced, the onus is on authors to make these comparisons if they're not available. Furthermore, EnerVerse with S-RGBD -> RGB with 1/2 renders is not provided in Table 3.
> > * **Impact of GS:** What are the additional analyses that are currently available studying the impact of GS? From L123 - 137, 4DGS is highlighted as an important part of the data flywheel. Not understanding the impact of this through experimental analyses is concerning.

---

> > > ### Author Response · Authors · 2025-08-07
> > >
> > > We sincerely thank the reviewer for their thoughtful feedback, acknowledgment of our clarifications, and recognition of the additional experiments we conducted. We appreciate the opportunity to further address the remaining concerns, and we are committed to providing detailed responses and additional analyses.
> > >
> > > >  I'm also wondering what the paper would look like after all these clarifications are incorporated, ... this has been noted by reviewer hqTx as well.
> > >
> > > Thank you for your comment. We would like to clarify that the revisions to the paper are primarily focused on improving the clarity of the presentation and incorporating additional experiments to provide a more comprehensive analysis. Importantly, these changes do not involve significant modifications to the core methodology, structure, or contributions of the paper.
> > >
> > > The paper’s main structure and experimental logic are already established. The clarifications we are making aim to address specific questions raised during the review process, ensuring that the paper is as clear and accessible as possible. These revisions are refinements rather than fundamental changes, and the original content and contributions remain intact.
> > >
> > > Additionally, we appreciate that, following the rebuttal discussion, **Reviewer hqTx** has expressed support for the paper, stating: **“I increased my score and I am in favor of accepting this paper.”** We believe this reinforces that the paper is already in a strong position, with the clarifications and additional experiments further enhancing its quality without altering its core contributions.
> > >
> > > Finally, we commit to producing a high-quality revised version that fully integrates all clarifications and improvements, ensuring that the final submission is polished and complete. We hope this alleviates any concerns about the stability of the paper or its readiness for publication.
> > >
> > >
> > > > 4D Model details
> > >
> > > Thank you for pointing this out and for sharing your concern. We appreciate your suggestion regarding providing a more thorough, from-the-scratch explanation of the architecture. We will make sure to incorporate a more detailed description and corresponding illustrations in a future version of the paper to address this issue and improve its clarity and accessibility for readers.
> > >
> > > > Thanks for the clarification about latency requirements. Please add this to the paper.
> > >
> > > Thank you for acknowledging this, we will add this into the revised version.
> > >
> > > > What is EnerVerse-LTX? Why is there no comparison to pure LTX without any of the contributions from this paper?
> > >
> > > Thank you for your question. As discussed in our previous response, we replaced the original DynamiCrafter (DC) video backbone with the LTX video backbone, while keeping other architectural designs identical.
> > >
> > > Compared to DynamiCrafter,  LTX operates in a more compact latent space and incorporates a well-designed spatio-temporal self-attention mechanism. Additionally, the LTX authors constructed a larger-scale dataset to support their method. Thanks to these advances, the EnerVerse-LTX variant demonstrates improved performance under the same training budget.
> > >
> > > However, due to **time and computational constraints** during the first-round discussion period, we were unable to complete the full training of all variants.  In this round discussion, we managed to provide the complete results. However, both LTX variants were trained for only ~30k steps, which is fewer than the fully trained DynamiCrafter setup due to the time limitations.  Since LTX is originally designed for long-video generation, we did not apply the FreeNoise mechanism to LTX-based models.
> > >
> > > We now provide the complete metrics in the table below:
> > >
> > > | Method            | PSNR (↑) | FVD (↓)   |
> > > |--------------------|----------|-----------|
> > > | DC-FN             | 25.42    | 445.95    |
> > > | LTX               | 25.92    | 550.85    |
> > > | EnerVerse-DC      | 26.10    | 404.65    |
> > > | EnerVerse-LTX     | 27.80 | **400.55**|
> > >
> > > From the table, we observe that EnerVerse-LTX achieves significant improvement in the FVD metric compared to LTX, highlighting the clear advantage of our design for long-sequence modeling.

---

> > > ### Author Response · Authors · 2025-08-07
> > >
> > > > If this is not the assumption, then what are the alternative ways to evaluate video generation models in terms of their fit for robotics control.
> > >
> > > Thank you for your follow-up question. As we stated in our first-round discussion, we do not assume a naïve one-to-one mapping whereby any uplift in general video-generation metrics (e.g., FVD, SSIM) will automatically yield better robotic control. Rather, our claim is that representational alignment—adapting the video model’s latent space to encode 3D, action-conditioned dynamics—is key to bringing gains in downstream robotic tasks.
> > >
> > > To expand further, we observe that our model's action predictions strictly follow the trajectories depicted in the generated videos. This suggests that the plausibility of the actions and the realism of robot-environment interactions in the generated videos are critical factors. However, image quality (e.g., sharpness or resolution) does not necessarily have a linear correlation with action prediction quality. For instance, reducing the number of denoising steps during generation (resulting in noisier images) or lowering the video resolution does not significantly degrade the quality of the predicted actions.
> > >
> > > To support this, we conducted tests on LIBERO with video diffusion denoising steps set to 1, 5, and 10, observing that the final policy success rate remains virtually unchanged. This highlights that for architectures like ours, which rely on generation-guided action planning, traditional video-generation benchmarks (which focus on image quality) are not adequate. Instead, an Embodied World Model benchmark is necessary to properly evaluate the alignment of video generation with robotic tasks. Recent work like EWMBench provides a promising direction for such evaluations.
> > >
> > > That said, we agree that producing high-quality videos aligned with the actions is still beneficial, especially as a data engine for training future world models. As Reviewer hqTx pointed out, action-aligned video generation can pave the way for better data generation pipelines.
> > >
> > > On the flip side, if the generated videos are completely chaotic or nonsensical, it would indeed indicate issues with the architecture. For example, in L259 Table 5:
> > >
> > > When no video generation model weights are loaded, policy learning completely fails, as the system lacks meaningful visual representations. Even when pre-trained video generation weights are loaded but no visual generation adaptation is performed, policy learning suffers significantly. From this perspective, video generation quality does have some relevance to robotic task performance, but the relationship is not direct or absolute.
> > >
> > > In conclusion, for embodied world models like ours, a new evaluation metric beyond traditional video generation benchmarks is urgently needed. This represents an important direction for future research.
> > >
> > > > Baselines across benchmarks.
> > >
> > > Thank you for bringing this up. Among the currently available open-sourced baselines, **OpenVLA** is one of the most widely used and community-recognized frameworks. To address concerns about consistency, we followed the official guidelines for OpenVLA and fine-tuned it on the **CALVIN benchmark**, providing the results below.
> > >
> > > In both **CALVIN** and **LIBERO**, we use the official evaluation metrics—**Success Rate (SR)** for LIBERO and **Task Completion Step Length (Len)** for CALVIN. As shown in the table, our method consistently outperforms the OpenVLA baseline across both benchmarks, demonstrating the generality and effectiveness of our approach. We also confirm that the performance of OpenVLA on CALVIN reported here aligns with other publicly available results, such as [1].
> > >
> > > | Method   | Benchmark | Visual Input | Metric | Performance |
> > > |----------|-----------|--------------|--------|-------------|
> > > | OpenVLA  | LIBERO    | S-RGB        | SR     | 76.5        |
> > > | Ours     | LIBERO    | S-RGB        | SR     | **84.1**    |
> > > | OpenVLA  | CALVIN    | S-RGB        | Len    | 0.95        |
> > > | Ours     | CALVIN    | S-RGB        | Len    | **3.00**    |
> > >
> > > These results shows that our method achieves superior performance on both benchmarks.
> > >
> > > [1] Unveiling the Potential of Vision-Language-Action Models with Open-Ended Multimodal Instructions

---

> > > ### Author Response · Authors · 2025-08-07
> > >
> > > >  EnerVerse with S-RGBD -> RGB with 1/2 renders is not provided in CALVIN.
> > >
> > > Thank you for pointing this out. We would like to provide clarification:
> > >
> > > - Differences in Benchmarks and Simulators: The CALVIN benchmark is relatively older than LIBERO and uses a different underlying simulator. This makes it significantly more challenging to add new cameras and render data with updated camera settings, as modifications to the simulator are not straightforward. However, we need these data during our training stage.
> > >
> > > - Fair Comparisons Across Baselines: To the best of our knowledge, no existing video-language-action (VLA) baselines on CALVIN utilize more than one third-person view. To ensure fair comparisons with prior work, we opted not to introduce additional camera views in CALVIN. Instead, for LIBERO, we explicitly report comparisons against MAIL using two S-RGB camera views, as LIBERO allows for a more flexible and modern setup.
> > >
> > > We acknowledge the importance of providing results for a multi-camera dataset. We will do our best to re-render the dataset with multiple camera views and include the corresponding results in a revised version of the paper. These re-render dataset and the script will be open-sourced with the community.
> > >
> > > > What are the additional analyses that are currently available studying the impact of GS?
> > >
> > > Thank you for your question. We have provided some initial visualizations in the Appendix (Fig 14). Additionally, we conducted qualitative analyses and observed that 4DGS enhances cross-view geometric consistency and dramatically reduces diffusion-induced hallucinations, particularly in challenging scenarios with robotic arm-object occlusions.
> > >
> > > For example, consider a robotic arm manipulating objects like gears or boxes on a table. With an initial setup of one third-view camera, there are timestamps where the arm occludes parts of the objects (e.g., during self-occlusion while manipulating a box). Over several chunks of diffusion generalization, if geometric constraints from 4DGS are not applied, the occluded object is prone to being hallucinated as a different type, even when a designed memory mechanism is used. In contrast, when geometric optimization across frames from 4DGS is applied, such hallucinations are significantly reduced, maintaining the object's identity and consistency.
> > >
> > > We will enrich the revised manuscript with additional visualizations of 4DGS’s effects and provide deeper analyses to further illustrate its impact.

---

> ### Comment · Reviewer_C5W7 · 2025-08-09
> **Updated reviewer response**
>
> I thank the authors for the follow-up experiments and responses. I have three remaining concerns:
> * paper requires drastic presentation changes
> * no comparisons between 3D video vs. 4D space generation for robotics
> * lack of quantitative ablations for 4DGS
>
> I've detailed the points below. As of this moment, I'm willing to raise my rating to **borderline reject**. However, I remain firm in my opinion that this paper needs significant presentation changes (and some further experimental changes) for acceptance.
>
> # Addressed concerns
>
> ## What is EnerVerse-LTX? Why is there no comparison to pure LTX without any of the contributions from this paper?
>
> The new updated LTX results are helpful. This concern is addressed. Thank you.
>
> ---
>
> ## CALVIN vs. LIBERO mismatch in baselines
> I appreciate the new results of OpenVLA on CALVIN. Thanks. This concern is addressed.
>
> ---
>
> # Remaining concerns
>
> ## Paper requires drastic presentation changes
> **TL; DR --- What would the paper look like after incorporating feedback? Not sure**
>
> I appreciate that the authors have agreed to fix the clarity concerns. While the technical details of the approach may remain the same, the presentation will have to be significantly different compared the original submission. For example, for some papers, authors could address this by saying "subsection XYZ will be rewritten as follows ....". However, here the entire Section 2 needs to be rewritten, potentially expanding the space requirement in the paper by 1-2 pages. That would require altering the remaining sections of the paper to accommodate this increased spacing requirement. I'm not comfortable accepting any paper without having a clear picture of what exactly the contents of that paper would look like.
>
> ---
>
> ## No comparisons between 3D video vs. 4D space generation for robotics
> **TL; DR --- What is the value of the core contribution of 4D space generation? How much does it help robotics?**
>
> I understand and agree with the authors arguments about better video not implying better robotic performance. However, that was not my follow up question. My question was "why not compare DC-FN and LTX on the robotic control tasks directly?". I'm looking for evidence substantiating the motivation for 4D embodied space generation (L114 - 116) - "Single-view approaches face ... extend the diffusion generator ... to multi-view video generation pipeline".
>
> *EnerVerse improves over the base generative models.* - Great. This is well demonstrated.
>
> *EnerVerse outperforms existing robotic baselines.* - Great. This is well demonstrated too.
>
> But how do the base generative models (without multi-view extensions from EnerVerse) fare on robotics? It's possible I'm mistaken, but is it not technically possible to add a policy head on top of the base video generative models (without multiple views) and train a policy?
>
> ---
>
> ## Lack of quantitative ablations for 4DGS
> **TL; DR --- What is the quantitative benefit of 4DGS?**
>
> The authors have again pointed to qualitative examples, which I do appreciate. However, my question was again not addressed. Specifically, "Not understanding the impact of this through **experimental analyses** is concerning.". This is a nice technical aspect of the generation pipeline, and my motivation for these questions is to better highlight the value through **quantitative analyses**, not simply through a few qualitative figures.

---

> > ### Author Response · Authors · 2025-08-09
> >
> > Thank you for the thoughtful follow-up. We appreciate that the LTX baseline concern and the CALVIN/LIBERO baseline alignment are now addressed; we will integrate these clarifications and numbers into the revised paper and appendix.
> >
> > > Paper requires drastic presentation changes
> >
> > Based on the discussion result, we agree on the need for clarity. However, the remaining items are primarily about presentation and supplementary details, most of which belong in the appendix (e.g., additional baselines, hyper-paramters, and implementation details). For some notions in the main paper, we do not think they will change the structure a lot. These do not alter the core method. We are prepared to move concise, high-value clarifications into the main text and to include the already-completed additional experiments and discussions to improve readability, without changing the paper’s claims. For this reason, we respectfully disagree that a drastic revision is required.
> >
> > > No comparisons between 3D video vs. 4D space generation for robotics.
> >
> > Thank you for the insightful follow-up. We agree that it is technically feasible to attach a policy head directly to base single-view video generators. To address your question, we provided additional comparison between base generative models (without multi-view/4D extensions) to EnerVerse.
> >
> > ### Direct Comparison: Base Video Generator vs. EnerVerse
> > **Setup**:
> > As described in L259 Table 5, we attached the same diffusion policy head to the **DynamiCrafter** (a single-view video generator) and trained it under the same schedule as EnerVerse-A. In contrast, EnerVerse-A, which has a similar architecture but is extended to accept multiple views, underwent **multi-view generation pretraining** before being fine-tuned with policy loss. Both models were evaluated on **LIBERO-Spatial** using single RGB (S-RGB) input.
> >
> > **Results**:
> > The base video model + policy head underperforms significantly compared to the multi-view/4D variant (EnerVerse-A). For full comparsion, please refer to L259.
> >
> > | Model                               | Multi-view / 4D extensions | Success Rate (LIBERO-Spatial) |
> > |-------------------------------------|----------------------------|-------------------------------|
> > | DynamiCrafter + diffusion policy    | No                         | 79.0                          |
> > | EnerVerse-A (ours)                  | Yes                        | **92.1**                      |
> >
> > These results highlight the benefits of multi-view/4D extensions in robotics tasks.
> >
> > We hypothesize that the **cross-view consistency** learned during multi-view pretraining provides  stronger geometric priors, which help the model better understand spatial relationships and occlusions.
> >
> > As further evidence, when testing EnerVerse with single-view inputs, the performance improves with additional rendered images (e.g., depth warping with provided S-RGBD). For example:
> > - On the LIBERO benchmark, EnerVerse achieves **84.1 (SR)** with single S-RGB input.
> > - Adding one rendered image improves performance to **85.5 (SR)**.
> > - Adding two rendered images maintains **85.5 (SR)** (see Table 2, L221).
> >
> > This demonstrates that multi-view extensions are necessary: The ability to incorporate additional rendered views at test time further enhances performance, showcasing the practicality and effectiveness of our multi-view/4D approach.

---

> > ### Author Response · Authors · 2025-08-09
> >
> > > Lack of quantitative ablations for 4DGS
> >
> > We thank the reviewer for pointing this out. To address this, we conducted additional experiments on the "arrange workpieces" task, where a robotic arm manipulates objects such as gears or boxes on a table. We chose this task because it contains timestamps where the arm occludes parts of the objects (e.g., during self-occlusion while manipulating a box).  Follow the data flywheel setting, given the task desciption, and one complete video from the head camera, we are aiming to generate the video from another view.
> >
> > We generated 30 episodes under the following two settings:
> > 1. **Without 4DGS**: Directly running the video generation pipeline in **EnerVerse-D**, without applying 4DGS.
> > 2. **With 4DGS**: Using **EnerVerse-D** to generate an initial video prediction, followed by 4DGS pipeline including rendering videos from 4DGS for target views, applying noise, and feeding the results back into EnerVerse for further refinement.
> >
> > For evaluation, we asked two human experts to assess the generated videos specifically for **diffusion-induced hallucinations**. The evaluators were blinded to the source of each video (i.e., whether 4DGS was applied).
> > | 4DGS       | Hallucination Count | Rate |
> > |------------|----------------------|------|
> > | Without    | 10                   | 0.33 |
> > | With       | 6                    | 0.20 |
> >
> > As shown in the table, 4DGS reduces hallucinations by 40% compared to the baseline (without 4DGS) for scenarios involving **self-occlusions**, where the robotic arm partially blocks objects during manipulation. These results quantitatively demonstrate the value of 4DGS in mitigating hallucinations.
> >
> > We acknowledge the need for more comprehensive quantitative analyses. To further support the benefits of 4DGS, we plan to conduct additional experiments on other tasks and metrics in the revised version of our paper.

---

### Note · Authors · 2025-08-13

We received valuable feedback from four reviewers with diverse perspectives and express our sincere gratitude for their insightful suggestions. We also thank the area chair for the thoughtful coordination.

## Recognized Contributions
- Strong empirical performance on LIBERO/CALVIN benchmarks and comprehensive real-world validation (multiple reviewers), complemented by thorough ablation studies (C5W7).
- Novel sparse memory mechanism, multi-view chunk-wise diffusion design, and effective integration of video generation with policy learning (hqTx, y6Qo, J1q3).
- Data engine design combining 4DGS with generative modeling (y6Qo) and the representational alignment approach (J1q3).


## Key Clarification
**Methodological Clarifications & Expanded Baselines**: We provided comprehensive implementation details (hyperparameters, dataset composition, etc.) and clarified the EnerVerse-D pipeline, addressing methodological concerns (C5W7, hqTx). Additional baseline comparisons demonstrate consistent advantages through cross-backbone validation with LTX and cross-benchmark evaluation with OpenVLA (C5W7). We also clarified computational latency during action policy inference (y6Qo).

**Component Ablations & Evaluations**: We quantified the value of multi-view extensions through comparisons with single-view baselines on LIBERO-Spatial (C5W7). New 4DGS ablations on occluded manipulation tasks demonstrate a 40% reduction in hallucination(C5W7, y6Qo). Additionally, we provided evidence that policy success rates remain stable across video denoising steps, supporting our discussion on representational alignment (J1q3, C5W7).

## Discussion Outcomes
Reviewer J1q3 emphasized our representational alignment and strong empirical evidence. Reviewer y6Qo stated that our rebuttal "strengthens confidence in the paper's technical validity, empirical thoroughness, and contribution significance." Reviewer hqTx increased their score and expressed being "in favor of accepting this paper" per discussion. Reviewer C5W7 acknowledged that "methodological details were largely clarified" and experimental concerns were "partially clarified."

## Closing
The review process significantly strengthened our work. With reviewers recognizing our novel sparse memory mechanism, multi-view world modeling, and strong empirical contributions, we are committed to incorporating all clarifications into a revised version that maintains the core contributions while improving presentation clarity.

---

### Decision · Program_Chairs · 2025-09-17

**Decision:**

Accept (poster)

**Comment:**

The paper initially received mixed reviews. Two reviewers leaned toward rejection while two supported acceptance. The reviewers agreed that the sparse memory mechanism is novel, the multi-view video generation approach improves performance, and the empirical results are strong, covering multiple benchmarks and real-world experiments. At the same time, some reviewers felt the writing was unclear and lacked important experimental details. They also noted that the method is complex, computationally inefficient, and may not scale well. In addition, comparisons with other video generation methods were missing, and the level of technical novelty was questioned.

In their rebuttal, the authors addressed most of these concerns, leading all reviewers to support acceptance. Some reviewers, however, remained concerned that significant revisions would be needed to integrate the rebuttal into the final version, and that the method’s complexity and reproducibility remain unresolved issues.

The AC agrees with the reviewers’ consensus and finds that the paper makes a valuable contribution, supported by strong empirical results that justify acceptance. At the same time, the AC emphasizes the importance of addressing the remaining concerns, and asks the authors to carefully integrate their rebuttal into the final version and provide sufficient details, including releasing code if possible, to ensure reproducibility.